# Double-strand breaks in facultative heterochromatin require specific movements and chromatin changes for efficient repair

Marieke R. Wensveen [1], Aditya A. Dixit[1], Robin van Schendel [2], Apfrida Kendek[1], Jan-Paul Lambooij[1], Marcel Tijsterman [2], Serafin U. Colmenares [3] & Aniek Janssen [1] ✉

DNA double-strand breaks (DSBs) must be properly repaired within diverse chromatin domains to maintain genome stability. Whereas euchromatin has an open structure and is associated with transcription, facultative hetero-chromatin is essential to silence developmental genes and forms compact nuclear condensates, called polycomb bodies. Whether the specific chromatin properties of facultative heterochromatin require distinct DSB repair mechanisms remains unknown. Here, we integrate single DSB systems in euchromatin and facultative heterochromatin in *Drosophila melanogaster* and find that heterochromatic DSBs rapidly move outside polycomb bodies. These DSB movements coincide with a break-proximal reduction in the canonical heterochromatin mark histone H3 Lysine 27 trimethylation (H3K27me3). We demonstrate that DSB movement and loss of H3K27me3 at heterochromatic DSBs depend on the histone demethylase dUtx. Moreover, loss of dUtx spe-cifically disrupts completion of homologous recombination at heterochro-matic DSBs. We conclude that DSBs in facultative heterochromatin require dUtx-mediated loss of H3K27me3 to promote DSB movement and repair.

Eukaryotic cells are continuously exposed to factors that break or chemically alter DNA. One particularly dangerous type of DNA damage is a double-strand break (DSB), in which both strands of the DNA helix are severed. Improper repair of DSBs can directly lead to insertions, deletions and chromosomal rearrangements associated with disease development including cancer[1]. To overcome these detrimental out-comes, cells have evolved mechanisms to repair DSBs of which the two main pathways are Non-Homologous End Joining (NHEJ) and Homo-logous Recombination (HR). During NHEJ, the severed DNA ends undergo limited end processing and are directly ligated, which can result in small insertions or deletions (indels) at the break site and is

therefore considered error-prone[2]. HR, on the other hand, is usually more precise since it relies on a homologous template to repair the DSB. During HR, the broken DNA ends undergo 5′ to 3′ end resection, generating 3′ single-stranded DNA (ssDNA) overhangs. This overhang invades a homologous sequence on the sister chromatid or the homologous chromosome, which serves as a repair template. In addition to HR and NHEJ, several alternative DSB repair pathways exist, such as single-strand annealing (SSA) or microhomology mediated alternative end-joining (MMEJ), which are error-prone mechanisms that rely on end resection and use homologous sequences to align the broken ends[2]. The choice of DSB repair pathway depends on multiple

[1]Center for Molecular Medicine, University Medical Center Utrecht, Universiteitsweg 100, Utrecht, the Netherlands. [2]Human Genetics Department, Leiden University Medical Center, Leiden, the Netherlands. [3]Department of Molecular and Cell Biology, University of California Berkeley, Berkeley, USA. ✉ e-mail: a.janssen-2@umcutrecht.nl

aspects, including cell cycle phase, the sequence context of the surrounding DNA, as well as the pre-existing chromatin state[2-4].

The eukaryotic nucleus consists of a variety of chromatin domains each characterized by specific molecular and biophysical properties. Whereas euchromatin has an open chromatin structure with actively transcribed genes, heterochromatin is more condensed and transcriptionally inactive. One type of heterochromatin is facultative heterochromatin, which is essential to silence specific developmental genes. Facultative heterochromatin can cover large genomic distances (e.g. developmental genes such as Hox genes)[5] or regulatory regions (e.g. promoters)[6]. This type of heterochromatin is enriched for Histone H3 Lysine 27 trimethylation (H3K27me3), H2AK118 ubiquitination (in fruit flies, H2AK119ub in mammals) and polycomb group (PcG) proteins, and accumulates in nuclear foci, called polycomb bodies[7-9]. Polycomb bodies cluster PcG-bound transcriptionally repressed genomic regions to maintain correct silencing of developmental genes[9-13]. Although the DSB response in open, euchromatic regions has been extensively studied, the DSB repair response in facultative heterochromatin remains largely unknown.

In the past decade, it has become clear that the pre-existing chromatin state can directly influence the DSB repair response. For example, DSBs in actively transcribed regions are prone to clustering and repair by HR[14,15]. Moreover, DSBs in centromeres[16], nucleoli[17-19] and constitutive heterochromatin domains[16,20-22] have been found to move outside the respective domains to facilitate repair. Previous evidence suggests that DSB movements can also occur within the inactive X chromosome[23], which is a specific type of facultative heterochromatin enriched for both H3K27me3 and H3K9me3[24]. Irradiation of female human fibroblasts resulted in the specific exclusion of DSB repair proteins outside the inactive X chromosome, suggestive of DSB movement[23]. Moreover, decompaction of the inactive X upon laser irradiation has also been observed[25]. Nevertheless, live imaging of individual DSBs to precisely monitor their dynamics within facultative heterochromatin has never been performed. More importantly, the response of polycomb bodies to DSBs in a physiological, in vivo setting, and whether this chromatin environment facilitates movements of DSBs, remains unknown.

Various histone modifications have been identified to play a role in DSB repair in euchromatin[3,26]. Silencing histone modifications, including H3K27me3[27,28] and H3K9me2/3[29,30] have been described to be deposited at DSBs in euchromatin, resulting in local, transient heterochromatinization and transcriptional silencing. To restart transcription after DSB repair in euchromatin, active removal of H3K27me3 by the mammalian histone demethylase UTX (Ubiquitously transcribed Tetratricopeptide repeat on X chromosome) has been suggested to occur specifically in cancer cells, not healthy fibroblasts[31]. In contrast to the accumulation of silencing marks at euchromatic DSBs, we previously identified a loss of the constitutive heterochromatin mark H3K9me2/3 at DSBs within *Drosophila* constitutive heterochromatin[32,33]. These findings suggest that eu- and heterochromatin regions require differential changes in silencing histone modifications to repair their DSBs. Whether specific H3K27-modifying activities are needed to repair DSBs in H3K27me3-enriched facultative heterochromatin domains remains untested.

Here, we study the dynamic DSB response in facultative heterochromatin in vivo by integrating inducible single DSB systems[22,34] in euchromatin and facultative heterochromatin regions in the fruit fly *Drosophila melanogaster*. Using high-resolution live imaging, we find that the majority of DSBs in polycomb bodies rapidly move outside these domains. Moreover, we find that facultative heterochromatic DSBs specifically undergo a local decrease in the canonical heterochromatin histone mark H3K27me3, which is mediated by the histone demethylase dUtx. Early steps of HR can occur efficiently within polycomb bodies and are independent of dUtx, while dUtx is required for subsequent DSB movement and completion of HR. Together, our results reveal that DSBs in facultative heterochromatin move outside the compacted polycomb bodies to promote timely repair by HR.

## Results
### Development of a single DSB system in facultative heterochromatin
To study DSB repair in facultative heterochromatin in detail in animal tissue, and directly compare responses to euchromatin, we generated a new set of inducible single DSB systems in *Drosophila*. We integrated our previously established in vivo DR-*white* reporters[22,34] into three facultative heterochromatin regions and two euchromatin regions (Fig. 1A, B), generating five fly lines, each containing the DR-*white* reporter at a distinct location. Upon expression of I-SceI, a DSB is induced in the upstream *white* gene. DSB repair pathway usage can subsequently be determined by sequencing the resulting repair products; HR with the i*white* sequence (intra-chromosomal or inter-sister) will generate an intact upstream *white* gene, while imperfect NHEJ may generate small insertions and deletions (indels) at the cut site (Fig. 1A). The DR-*white* reporters have been previously well characterized to study DSB repair -pathway choice and -dynamics in euchromatin[34,35] as well as constitutive heterochromatin[22,33].

We performed ChIP-qPCR (Chromatin Immuno-Precipitation followed by quantitative PCR (qPCR)) for the canonical facultative heterochromatin histone modification H3K27me3 and confirmed enrichment of H3K27me3 at the three DR-*white* integrations in heterochromatin when compared to the two euchromatic insertions (Fig. 1C). Moreover, internal controls revealed strong specificity of the antibody used for H3K27me3 ChIP analysis (Supplementary Fig. 1A).

To allow timed DSB induction, we combined our DR-*white* systems with either a heat-shock inducible *I-SceI* transgene (*hsp70.I-SceI*, Fig. 1D)[34] or an *ecDHFR-I-SceI* transgene[22], which depends on the ligand trimethoprim to stabilize the ecDHFR-I-SceI protein (Fig. 1E). We first tested the efficiency and inducibility of our DR-*white* systems by performing ChIP-qPCR for phosphorylated H2Av on Serine 137 (γH2Av, γH2AX in mammals), one of the earliest chromatin markers of DSB induction[36]. Heat-shock inducible expression of I-SceI results in a local increase in γH2Av levels within six hours at both euchromatic and heterochromatic DR-*white* loci (Fig. 1F, G). Moreover, we find the appearance of single γH2Av foci in nuclei of imaginal discs six hours after heat-shock inducible I-SceI expression in larvae containing either a eu- or heterochromatic DR-*white* insertion (16–23% of cells contained a single γH2Av focus compared to 3–5% in control cells) (Supplementary Fig. 1B, C). These results suggest efficient DSB induction in both eu- and heterochromatic loci.

To directly determine which DSB repair pathways play a role in facultative heterochromatin, we performed Sanger sequencing followed by TIDE analysis[37] on the DR-*white* reporters upon I-SceI expression. Feeding trimethoprim throughout development (3–4 days) to DR-*white* larvae expressing ecDHFR-I-SceI (Fig. 1E) results in the appearance of repair products in both eu- and heterochromatin (Fig. 1H). Repair rates vary slightly between integration sites (55–70%), possibly reflecting differential cutting efficiency or repair timing at the different sites. However, no significant differences in the number of identified repair products were found between heterochromatin and euchromatin, indicating that both domains undergo efficient DSB induction and repair. More importantly, repair pathway analyses revealed that I-SceI-induced DSBs in both eu- and heterochromatin regions employ HR (17–26%) and NHEJ (74–83%) to a similar extent (Fig.1I). Ec-DHFR-I-SceI larvae are administered trimethoprim throughout all larval stages (1st through 3rd instar). Recent findings indicate that the choice of DSB repair pathway remains consistent between 1st and 3rd instar larvae[34,35]. This suggests that larval stage is unlikely to affect the DSB repair pathway choice using the ecDHFR-I-SceI system. Moreover, uninduced *hsp.I-SceI*/DR-*white* or *ecDHFR.I-SceI*/DR-*white* larvae exhibited low background levels of DSBs

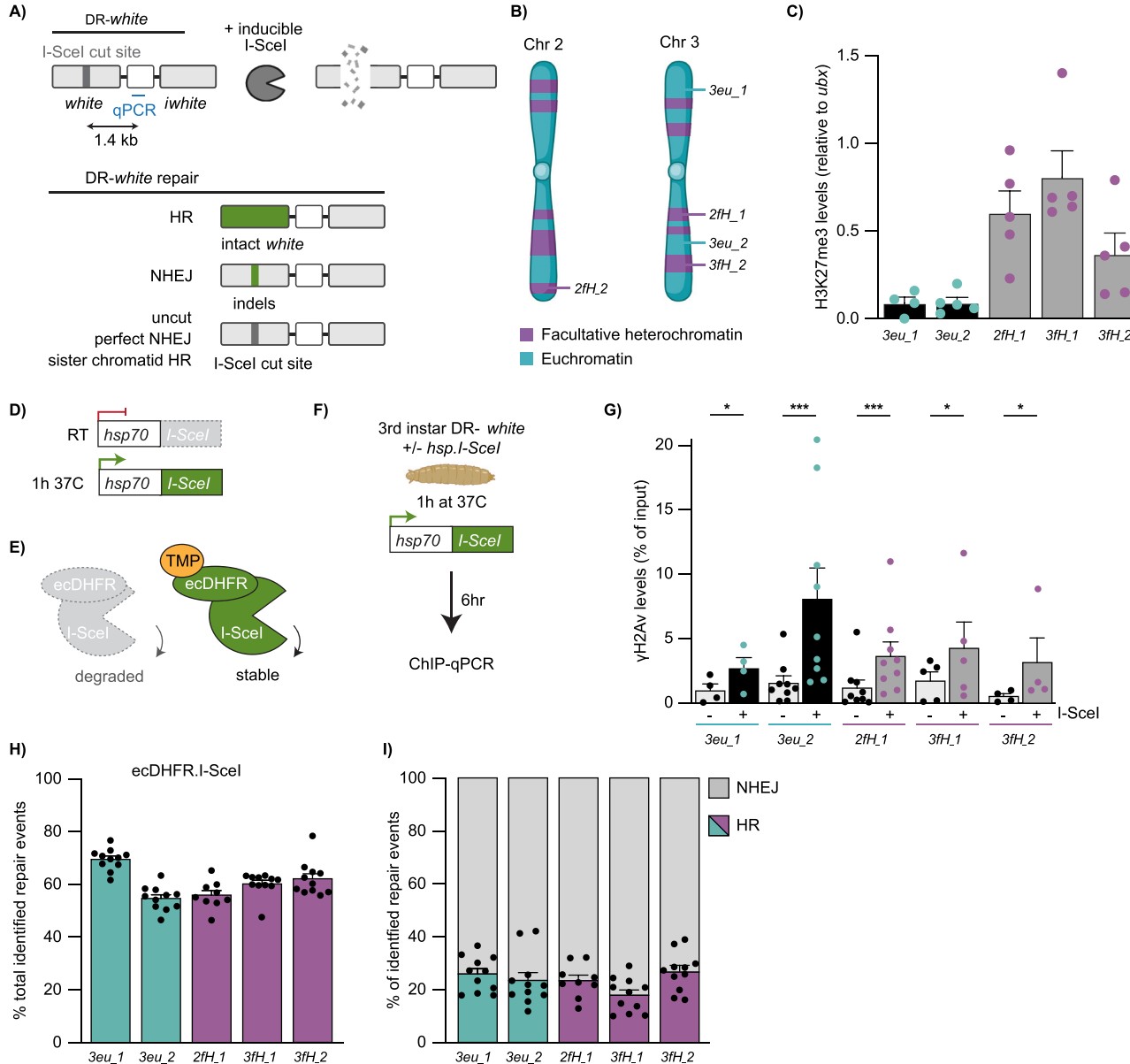

**Fig. 1 | DR-*white* system to induce single DSBs in euchromatin and facultative heterochromatin.** **A** Schematic of the DR-*white* system as previously established in[22]. I-SceI expression creates a DSB in the upstream *white* gene. Homologous recombination (HR) with the i*white* sequence (intra-chromosomally or with the sister chromatid) results in loss of the I-SceI cut site (−23bp). Imperfect non-homologous end joining (NHEJ) may result in indels at the DSB site. Perfect NHEJ or HR using the I-SceI sequence on the sister chromatid can recreate the I-SceI cut site. **B** Schematic of DR-*white* integration sites in five fly lines (2× in euchromatin [*3eu_1, 3_eu2*] and 3× in facultative heterochromatin [*2fH_1, 3fH_1, 3fH_2*]). Created in BioRender. Janssen, A. (2021) BioRender.com/h58g617. **C** ChIP-qPCR analysis for H3K27me3 at the DR-*white* locus (primers Fig. 1A). H3K27me3 levels were normalized using *ubx* gene primers (*ubx* has high H3K27me3 levels[63]). Averages +SEM are shown for *n* = 5 biological replicates (except *3eu_1*: *n* = 4). p-values: 3eu_1 = 0.0350, 3eu_2 = 0.0009, 2fH_1 = 0.0006, 3fH_1 = 0.0333, 3fH_2 = 0.0432. **D** Schematic of the *hsp.I-SceI* construct. The *hsp70* promoter upstream of *I-SceI* can be activated by shifting larvae for one hour (1 h) to 37 °C. RT=room temperature. **E** Schematic of the ecDHFR-I-SceI system. Proteasomal degradation of ecDHFR-I-SceI can be blocked by adding trimethoprim (TMP). **F** Experimental set up for ChIP-qPCR experiments. 3rd instar DR-*white* larvae with and without *hsp.I-SceI* (control) were heat-shocked. Six hours later chromatin was subjected to ChIP-qPCR (primers Fig. 1A). Figure partly created in BioRender. Janssen, A. (2024) BioRender.com/n98i722. **G** ChIP-qPCR analysis for yH2Av in the absence (-) and presence (+) of hsp.I-SceI (as in F). Averages +SEM are shown for *n* = 4 (*3eu_1, 3fH_2*), *n* = 5 (*3fH_1*), or *n* = 9 (*3eu_2, 2fH_1*) biological replicates. **H**, **I** DR-*white*/ecDHFR-I-SceI larvae were fed TMP for 3-4 days. Repair products at the upstream *white* gene were amplified and analyzed using TIDE[37]. Graphs show the total identified repair events in all PCR products (**H**) and the percentage of repair products with either HR (color) or NHEJ (gray) signatures (**I**). Bars indicate averages +SEM of *n* = 11 single larvae (biological replicates) (except *2fH_1*: *n* = 9). If not shown, p-value (>0.05), (*) p-value ≤ 0.05, (***) p-value ≤ 0.001, ratio two-sided paired t-test (**G**), one-way ANOVA + Tukey's multiple comparisons test (**I**). Source data are provided as Source Data file.

(Supplementary Fig. 1C, D), indicating limited leakiness of the I-SceI systems in the absence of induction.

In line with our results using the ecDHFR-I-SceI system, expression of I-SceI using the *hsp.I-SceI* transgene also resulted in the appearance of DSB repair products (7–19%) and similar HR and NHEJ repair percentages in eu- and heterochromatin DR-*white* integrations (Supplementary Fig. 1E, F), suggesting that eu- and facultative heterochromatin DSBs do not show differences in repair pathway choice using the DR-*white* reporters. Finally, our heat-shock inducible system resulted in an overall lower number of repair products when compared

to ecDHFR-I-SceI, likely reflecting the shorter duration of I-SceI expression (24 h using the *hsp.I-SceI* system compared to 3–4 days in the *ecDHFR-I-SceI* system).

To confirm the role of HR and NHEJ repair at facultative heterochromatic DSBs, we depleted several DSB repair proteins using RNAi and analyzed DR-*white* repair products following heat shock−inducible I-SceI expression (Supplementary Fig. 2). Loss of the HR repair proteins DmCtIP (Supplementary Fig. 2A, B) and DmRad51 (Supplementary Fig. 2E) resulted in a decrease in HR repair products, indicating that the identified HR repair products indeed reflect end-resection dependent HR repair. Additionally, we depleted the NHEJ protein DmKu70 using RNAi (Supplementary Fig. 2C, D). As expected, loss of DmKu70 resulted in an increase in HR repair products (47–57% compared to 18–19% in control) after I-SceI induction (Supplementary Fig. 2D), confirming the role of NHEJ repair at DSBs in facultative heterochromatin.

Finally, we performed Illumina sequencing of DR-*white* repair junctions to identify whether alternative DSB repair pathways are used in facultative heterochromatin (Supplementary Fig. 3). Illumina sequencing identified 58–75% of repair products at DR-*white* loci following ecDHFR-I-SceI induction (Supplementary Fig. 3A). A total of 15–25% of these repair products were repaired by HR and ~72–80% was repaired by NHEJ, similar to what we identified by TIDE analyses (Supplementary Fig. 3B, Fig. 1H, I). The majority of all NHEJ products analyzed (99.0–99.4%) contained small deletions with the majority being 1 base pair deletions (Supplementary Fig. 3C, E). Further analysis of these deletion products revealed that 3–9% of these deletion products contained microhomologies of 2 to >4 base pairs, suggesting that microhomology mediated alternative end-joining (MMEJ) could play a minor role in DSB repair in facultative heterochromatin using this reporter (Supplementary Fig. 3B, D).

Together, these data reveal that our systems efficiently induce DSBs, with little variation between euchromatin and facultative heterochromatin regions in terms of repair products, suggesting that both chromatin regions undergo similar DSB repair efficiency and pathway choice. Our inducible single DSB system in vivo therefore allows us to perform detailed analyses of DSB repair in facultative heterochromatin, and directly compare it to the DSB response in euchromatin.

## DSBs rapidly move outside polycomb bodies

Specific DSB spatiotemporal dynamics are associated with a variety of chromatin domains, such as centromeres[16], nucleoli[17–19] and constitutive heterochromatin[16,20,22]. These dynamics include the movement of DSBs to the periphery of the respective domain[16,19–21]. Facultative heterochromatin forms distinct domains in the fly and mammalian nucleus, termed polycomb bodies[7–9]. We wished to determine whether the distinct molecular- and biophysical- properties of polycomb bodies[38–41] could impact DSB dynamics and promote movements similar to those previously identified in other nuclear domains. To this end, we employed our DR-*white* systems to perform in vivo live imaging of single DSBs in facultative heterochromatin (Fig. 2A) using fluorescently tagged Mu2 to visualize DSBs, and fluorescently tagged polyhomeotic-proximal (ph-p) to visualize polycomb bodies (Fig. 2B). Mu2 is the *Drosophila* ortholog of mammalian MDC1, and directly binds γH2Av[42,43], while ph-p is one of the four core subunits of *Drosophila* PRC1 (Polycomb Repressive Complex 1) and is enriched in fly polycomb bodies[44,45] (Fig. 2B). Strikingly, our live imaging analyses revealed that the majority of Mu2 foci (60%) that appear within polycomb bodies move outside the domain within ten minutes (Fig. 2C, D, Supplementary Fig. 4A), while euchromatic DSBs that appear outside polycomb bodies remain outside until their disappearance (Supplementary Fig. 4B). Importantly, these movements are not specific for I-SceI-induced DSBs, since inducing DSBs in larval tissue using 5 Gy γ-irradiation resulted in similar DSB dynamics in ph-p

marked polycomb bodies (Fig. 2E, F). The majority of radiation-induced Mu2 foci that appeared within polycomb bodies moved outside this domain within ten minutes after appearance, whereas euchromatic DSBs are resolved outside polycomb bodies (Supplementary Fig. 4C). By assessing the kinetics of Mu2 focus appearance (onset of DSB repair) and disappearance (resolution of repair focus), we find that 50% of the Mu2 foci in polycomb bodies disappear within 30–80 min depending on the DSB induction method (Supplementary Fig. 4D, E, I-SceI and IR respectively). Although Mu2 presence is not an absolute measure of repair timing due to the possible persistence of γH2Av and Mu2 after DSB repair is finished[46], these kinetics are in line with previous research studying I-SceI − dependent DSB repair in euchromatin and constitutive heterochromatin in flies[22], as well as DSB repair in mammalian cells using IR[47]. Taken together, our live imaging data demonstrate that DSBs move outside polycomb bodies to continue repair.

## H3K27me3 reduction at heterochromatic DSBs is mediated by the demethylase dUtx

Considering the compact and silent state of facultative heterochromatin, we hypothesized that local chromatin changes could coincide with the specific DSB movements in polycomb bodies. To this end, we assessed the levels of the canonical facultative heterochromatin histone modification H3K27me3 by ChIP-qPCR 1.4 kilobases from the DSB sites (Figs. 1A, F, 3A). We observed a decrease in H3K27me3 (loss of 22–34%) at two of the three heterochromatic DSB sites after I-SceI induction, while H3K27me3 at euchromatic DSB sites remained unchanged (Fig. 3A). Using a qPCR primer set further away (3.1 kilobases) from the DSB site, we find a similar reduction in H3K27me3 (Supplementary Fig. 5A). Since 16–23% of cells obtain a single DSB six hours after heat shock−inducible I-SceI expression (Supplementary Fig. 1B, C), this decrease in H3K27me3 levels suggests a significant loss of H3K27me3 at individual heterochromatic DSBs. To exclude that the observed reduction in H3K27me3 is due to histone loss at the break site, we performed ChIP-qPCR for histone H3, which did not reveal any significant differences in histone H3 levels at euchromatic and heterochromatic DSB sites (Fig. 3B). The reduction in H3K27me3 was observed at two of the three heterochromatic integrations, suggesting that not all heterochromatic DSBs induce evident loss of H3K27me3. Nevertheless, our data reveal that DSBs in facultative heterochromatin are frequently accompanied by a local reduction in H3K27me3.

We hypothesized that the reduction in H3K27me3 levels at heterochromatic DSBs could be mediated by a histone demethylase that actively removes the methyl groups from H3K27. In *Drosophila*, dUtx is the only protein described to demethylate H3K27me3[48,49]. To determine whether dUtx removes the methyl group at heterochromatic DSB, we depleted dUtx using RNAi in 3rd instar larvae (Fig. 3C) and assessed H3K27me3 levels at the DSB sites using ChIP-qPCR. Indeed, dUtx depletion leads to retention of H3K27me3 at DSBs in heterochromatin, whereas the H3K27me3 levels at euchromatic DSBs remain unaffected upon DSB induction in the presence or absence of dUtx (Fig. 3D). As an orthogonal approach to Chromatin IP at single DSB sites in vivo, we assessed H3K27me3 levels at DSBs in cell culture using a live imaging-based approach employing a fluorescent mintbody that binds H3K27me3[50,51] (Fig. 3E−G). Using this method, we identify a drop in H3K27me3 signal (~10%) at DSBs within polycomb bodies, indicating that DSBs already lose H3K27me3 before movement outside polycomb bodies. More importantly, in line with our ChIP analysis, we find that loss of dUtx prevents the reduction in H3K27me3-mintbody signal at DSB sites in facultative heterochromatin (Fig. 3G). Interestingly, we also find a reduction in the other canonical (facultative) heterochromatin mark H2AK118 ubiquitination (H2AK119ub in mammals) (Supplementary Fig. 5B), as well as a dUtx-dependent loss of ph-p, a PRC1 complex member (Supplementary Fig. 5C−E), at

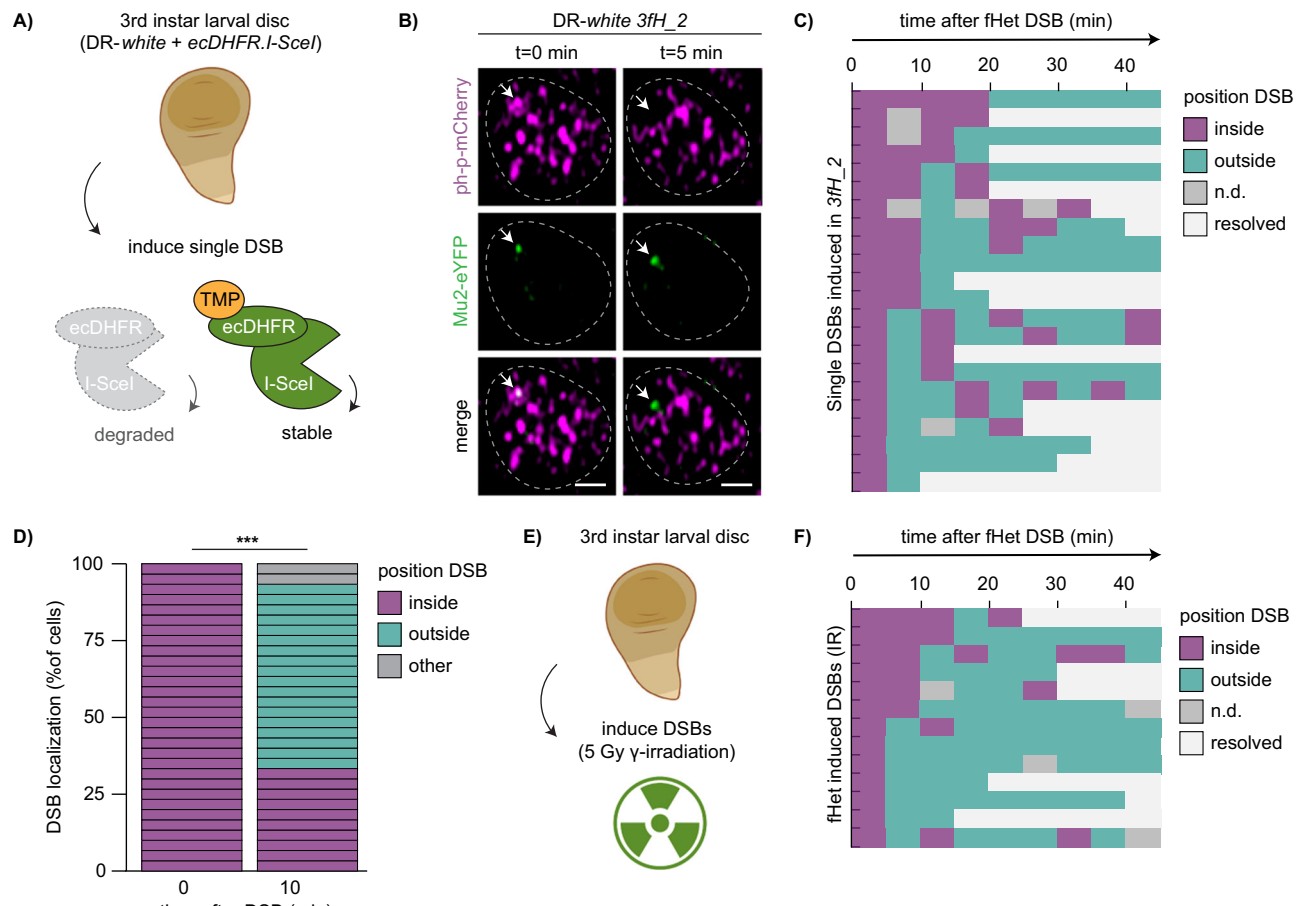

**Fig. 2 | DSBs in facultative heterochromatin move outside polycomb bodies.**
**A** Set up to analyze the dynamics of a single double-strand break (DSB) in larval tissue. Wing discs of DR-*white/ecDHFR-I-SceI* 3rd instar larvae were dissected and placed in medium with trimethoprim (TMP) to stabilize ecDHFR-I-SceI. Figure partly created in BioRender. Janssen, A. (2023) BioRender.com/b71q374.
**B** Representative live images of Mu2-eYFP focus dynamics (DSB marker, green) within the ph-p-mCherry domain (polycomb marker, magenta) in 3rd instar larval wing disc. Arrow indicates Mu2-eYFP focus arising within ph-p-mCherry domain (0 min) and moving outside this domain (5 min), min = minutes. Scale bar = 1 μm. Dotted lines outline nuclei. **C** Quantification of live imaging of Mu2-eYFP (DSB) dynamics and kinetics over time relative to the ph-p-mCherry domain using single DSB induction (ecDHFR-I-SceI, *3fH_2*). Each row indicates one single Mu2-eYFP focus that appeared (0 min) within a polycomb body. Mu2-eYFP foci were followed up to 45 min after appearance. Colors indicate localization of the Mu2-eYFP focus (inside polycomb body [purple], outside [green], not detectable [gray], or resolved [white]). **D** Quantification of Mu2-eYFP (DSB) localization 10 min after Mu2-eYFP appearance in ph-p domains (*2fH_1* and *3fH_2*). Colors indicate localization of the Mu2-eYFP focus (inside polycomb body [purple], outside [green], not detectable or resolved [gray]). *p*-value < 0.0001. **E** Set up to analyze the dynamics of DSBs using irradiation. Wing discs of 3rd instar larvae were dissected and placed in medium, followed by exposure to 5Gy gamma-irradiation (IR). Figure partly created in BioRender. Janssen, A. (2023) BioRender.com/b71q374, and Janssen, A. (2023) BioRender.com/o68s986. **F** Quantification of live imaging of Mu2-eYFP (DSB) dynamics and kinetics over time using 5 Gy IR (quantification as in **C**). (***) p-value ≤ 0.001, two-sided Chi-square test (**D**). Source data are provided as Source Data file.

heterochromatic DSBs. Together, our data suggest that multiple canonical heterochromatin components are lost at DSBs in facultative heterochromatin.

Loss of dUtx did not alter the levels of the DSB marker γH2Av, indicating that the retention of H3K27me3 at DR-*white* sites upon dUtx depletion is not due to inefficient cutting by I-SceI (Supplementary Fig. 5F). Additionally, as expected, DSBs in facultative heterochromatin do not affect levels of the constitutive heterochromatin mark H3K9me3, which are almost undetectable in facultative heterochromatin in undamaged as well as damaged situations (Supplementary Fig. 5G). Together, these data suggest that dUtx mediates the removal of H3K27me3 at DSB sites specifically in facultative heterochromatin.

### DSB movement and HR repair in facultative heterochromatin depend on dUtx

We next hypothesized that the local loss of H3K27me3 at heterochromatic DSBs could be required to promote DSB movement outside polycomb bodies. To test this, we turned to *Drosophila* cells in culture which allow for in-depth visualization of repair processes in combination with RNAi-mediated depletions. We visualized the dynamics of the early HR protein ATR Interacting Protein (ATRIP) within polycomb bodies (ph-p domains) in the presence or absence of dUtx using live imaging of 5 Gy γ-radiated *Drosophila* Kc cells (Fig. 4A). ATRIP binds to RPA-coated ssDNA overhangs, which are produced early in HR during 5' to 3' end resection of the DSB[52]. We find that upon irradiation of control cells, ATRIP foci appear inside polycomb bodies and move outside these domains within ten minutes (Fig. 4B, C, Supplementary Fig. 6A), which recapitulates our findings on Mu2 foci dynamics at heterochromatic DSBs in larval tissues (Fig. 2C, D, F). Interestingly, ATRIP gets recruited inside polycomb bodies upon irradiation in the absence of dUtx, suggesting that end resection steps are independent of dUtx (Fig. 4D, E, Supplementary Fig. 6B). However, dUtx depletion does lead to a delay in the movement of ATRIP-coated DSBs outside polycomb bodies. In control cells, 77% of ATRIP foci move outside the polycomb bodies within 10 min, while in dUtx depleted cells only 53%

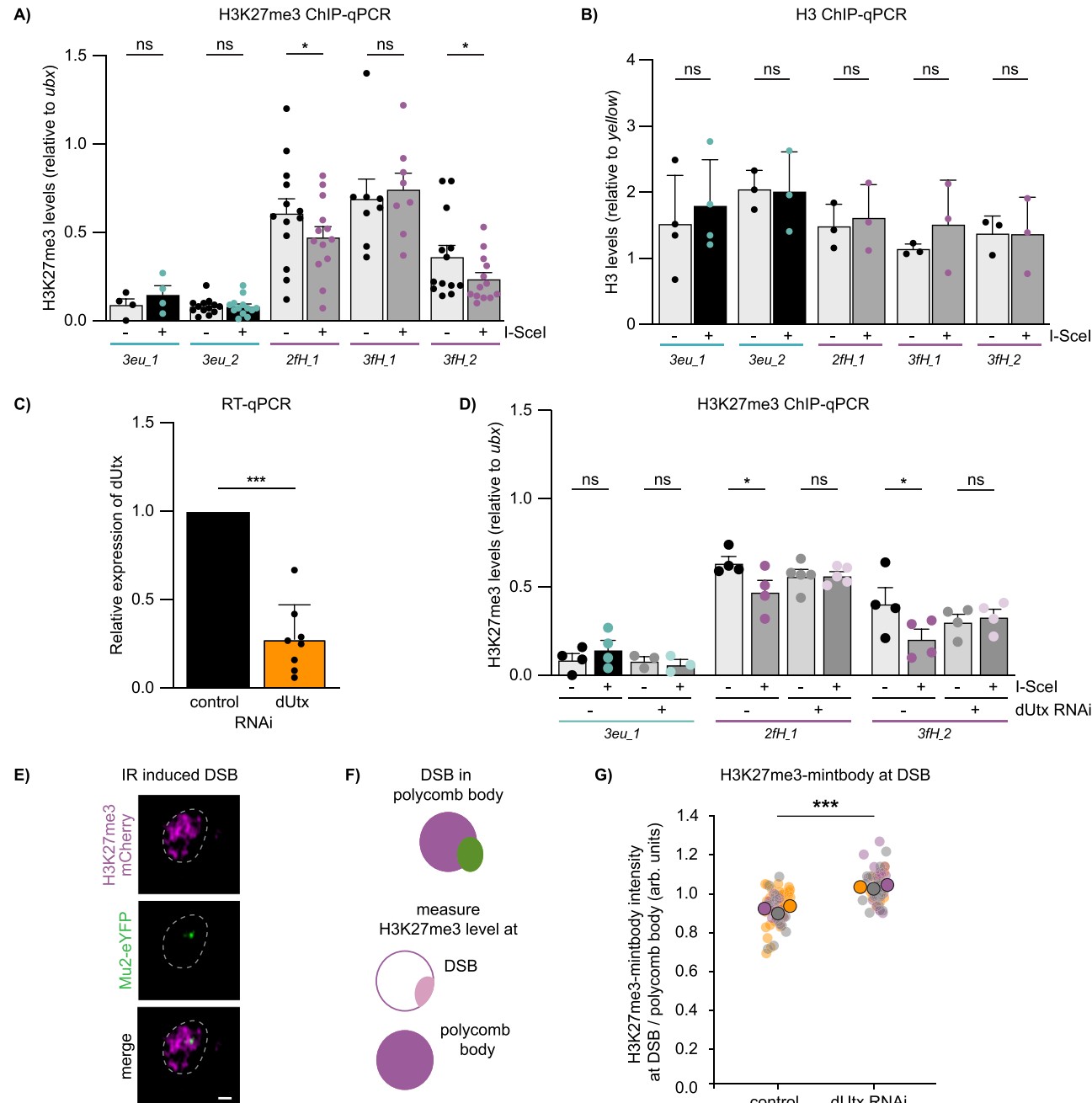

**Fig. 3 | Loss of H3K27me3 at facultative heterochromatic DSBs is mediated by dUtx. A, B** ChIP-qPCR analysis for H3K27me3 (**A**) and H3 (**B**) in the absence (-) and presence (+) of *hsp.I-SceI* (as in Fig. 1F) at indicated DR-*white* integration sites. H3K27me3 levels were normalized using a *ubx* qPCR primer set as an internal positive control and H3 levels using a *yellow* qPCR primer set. Averages +SEM are shown for *n* = 13 biological replicates (except *3eu_1*: *n* = 4 and *3fH_1*: n = 8) (**A**) and *n* = 3 biological replicates (except *3eu_1*: *n* = 4) (**B**). p-values: 2fH_1 = 0.0340, 3fH_2 = 0.0482. **C** Relative dUtx expression level in 3rd instar larvae determined using RT-qPCR and normalized to an internal control gene (tubulin). Bars indicate averages +SD of 8 single larvae (biological replicates) per condition (luciferase control RNAi or dUtx RNAi). p-value < 0.0001. **D** ChIP-qPCR analysis for H3K27me3 in the absence or presence of a DSB (-/+ hsp.I-SceI), with or without dUtx RNAi (as in Fig. 1F). H3K27me3 levels were normalized as in **A**. Averages +SEM are shown for *n* = 4 biological replicates (except *3eu_1* (dUtxRNAi): *n* = 3, *2fH_1* (dUtxRNAi): *n* = 5). *p*-values: 2fH_1 −dUtx RNAi=0.0216, 3fH_1 −dUtx RNAi = 0.0431. **E** Representative image of irradiation-induced Mu2-eYFP focus (DSB marker, green) within the H3K27me3-mintbody domain (polycomb marker, magenta) in S2 cells within 30 min after 5 Gy IR. Scale bar = 1 μm. Dotted line outlines nucleus. **F** Schematic of the quantification of H3K27me3 levels at DSBs in polycomb bodies. Relative enrichment of H3K27me3 at DSBs is calculated by dividing H3K27me3 levels at the DSB (light pink) by H3K27me3 levels at the complete polycomb body (magenta). **G** Quantification of relative enrichment of H3K27me3 levels at DSBs in polycomb bodies with or without dUtx depletion, within 30 min after IR (quantification as in **F**). Graph represents three biological replicates per condition. Each big circle (gray, purple, orange) represents the average intensity of one experiment. Small circles represent individual cells within one experiment, arb. units = arbitrary units. p-value: 0.0009. (ns) not significant, (*) *p*-value ≤ 0.05, (***) *p*-value ≤ 0.001, two-sided paired t-test (**A**–**D**) and two-sided unpaired t-test (**G**). Source data are provided as Source Data file.

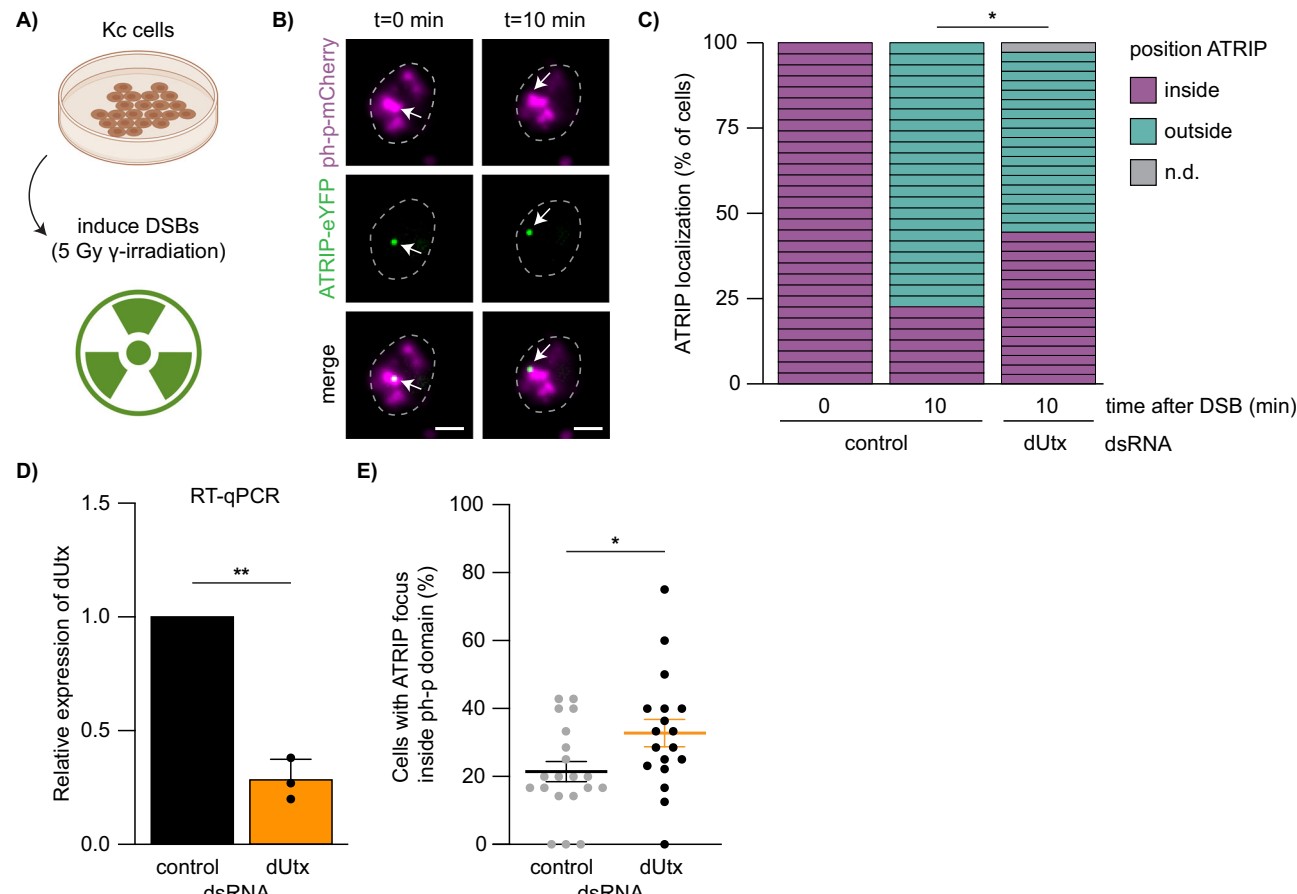

**Fig. 4 | DSB movement in facultative heterochromatin depends on dUtx. A** Cell culture set up to analyze the dynamics of ATRIP localization upon DSB induction using 5Gy gamma-irradiation. Figure partly created in BioRender. Janssen, A. (2023) BioRender.com/o68s986 and Janssen, A. (2023) BioRender.com/v72e853. **B** Representative images of ATRIP-eYFP focus (green) dynamics within the ph-p-mCherry domain (magenta). Arrow indicates ATRIP-eYFP focus arising within ph-p-mCherry domain (0 min) and moving outside this domain (10 min). Scale bar = 2 μm. Dotted lines outline nuclei. **C** Quantification of ATRIP-eYFP (DSB) localization 10 min after ATRIP-eYFP appearance, 3 biological replicates per condition (yellow control dsRNA or dUtx dsRNA). Colors indicate localization of the ATRIP-eYFP focus (inside polycomb body [purple], outside [green], not detectable or resolved [gray]). p-value: 0.0490. **D** Relative dUtx expression level in Drosophila Kc cells normalized to an internal control gene (tubulin), determined using RT-qPCR. Bars indicate averages +SD of 3 biological replicates per condition. p-value: 0.0053. **E** Quantification of the number of cells with an ATRIP-eYFP focus inside ph-p-mCherry domain divided by the total number of cells expressing both ATRIP-eYFP and ph-p-mCherry. Each dot indicates the percentage of cells with ATRIP foci overlapping with a ph-p domain per acquired image (total of 20 images (control) and 18 images (dUtx dsRNA) were aquired during 3 biological replicates). Averages +SEM are shown. p-value: 0.0289. (ns) not significant, (*) p-value ≤ 0.05, (**) p-value ≤ 0.01, two-sided Chi-square test (**C**), two-sided paired t-test (**D**) and two-sided unpaired t-test (**E**). Source data are provided as Source Data file.

move out within this timeframe (Fig. 4C, Supplementary Fig. 6B). In line with this, we find an increased accumulation of ATRIP foci within the ph-p domains (21% in control cells during the course of our imaging experiment, compared to 33% in dUtx-depleted cells) indicative of defects in DSB movement upon dUtx depletion (Fig. 4E). Together, these results suggest that the early steps of HR (end resection, ATRIP recruitment) occur efficiently within polycomb bodies and are independent of H3K27me3 demethylation at the DSB site. However, dUtx-mediated demethylation of H3K27me3 is required for the subsequent DSB movement outside the polycomb body.

To test whether this dUtx-dependent DSB movement is required for later repair steps in facultative heterochromatin, we employed our in vivo reporter system, which allows the direct assessment of DSB repair pathway choice (HR/NHEJ) by sequencing repair products (Fig. 1A). Strikingly, loss of dUtx in larvae revealed a 39–52% relative reduction in the proportion of HR repair products at two of the three heterochromatic DSB sites, while euchromatic DSB repair products remained unchanged (Fig. 5A). This reduction in HR repair was accompanied by an increase in NHEJ repair (Supplementary Fig. 7A). The effect on HR is not due to dUtx-dependent changes in DSB

induction efficiency as determined by γH2Av ChIP (Supplementary Fig. 5F). Moreover, effects are independent of the GAL4-driver used to drive dUtx dsRNA and is not due to indirect effects of the RNAi, since we can reproduce the reduction in HR using larvae mutant for dUtx and larvae expressing dUtx dsRNA under control of a daughterless-GAL4 (Supplementary Fig. 7B).

Although dUtx depletion did not affect relative HR levels at the *3fH_2* integration (Fig. 5A), we do find that loss of dUtx results in a reduction in the total number of identified repair products at the *3fH_2* integration, as well as at the *3fH_1* integration site (Supplementary Fig. 7C). A reduction in identified repair products is indicative of defects or delays in DSB repair. Interestingly, this reduction in total identified repair products is more evident when strictly assessing repair of the *3fH_2* region in wing disc tissues, which have strong silencing of genes nearby this integration site (i.e. hmx)[53] (Supplementary Fig. 7D). This indicates that DSB sites with high H3K27me3 levels depend more heavily on dUtx for repair. Indeed, brain tissues with high gene expression levels (low H3K27me3) nearby *3fH_2* do not show this reduction in repair efficiency upon dUtx depletion (Supplementary Fig. 7D). Together,

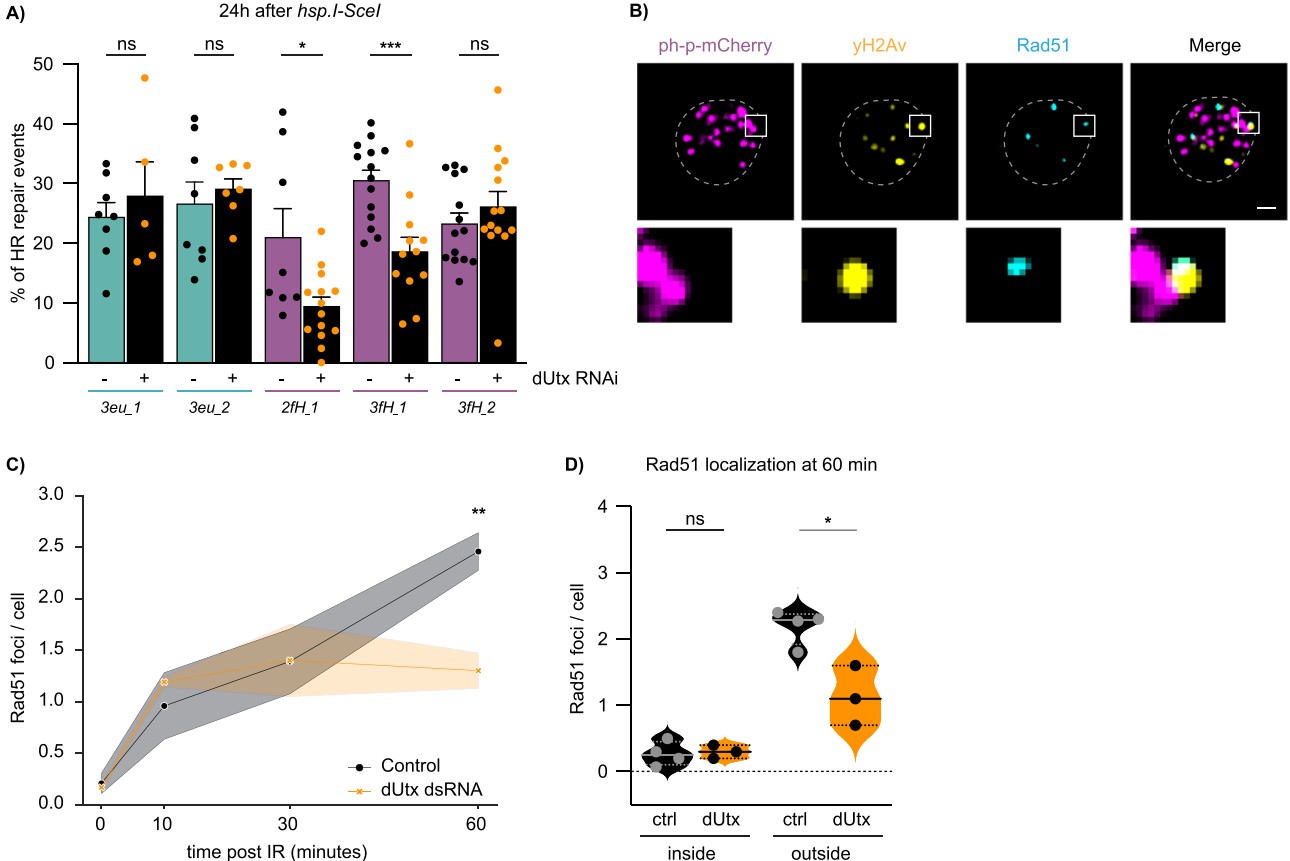

**Fig. 5 | dUtx is required for homologous recombination in facultative heterochromatin. A** DR-*white/hsp.I-SceI* larvae with (+) or without (-) dUtx RNAi were heat-shocked for one hour to induce I-SceI. Repair PCR products were Sanger sequenced 24 h after I-SceI induction and analyzed using the TIDE algorithm[37]. Graph shows percentage of identified homologous recombination (HR) products. Bars indicate averages +SEM of *n* = 14 single larvae (biological replicates) per condition (except *3eu_1*, *3eu_2*, *2fH_1*: *n* = 8, *3eu_1* (dUtx RNAi): *n* = 5, *3eu_2* (dUtx RNAi): *n* = 7, and *3fH_1* (dUtx RNAi): n = 12). p-values: 2fH_1 = 0.0122, 3fH_1 = 0.0005. **B** Representative image of a nucleus 60 min after 5 Gy irradiation immunostained for anti-γH2Av (yellow), anti-Rad51 (cyan) and anti-mCherry to visualize ph-p-mCherry (magenta). Zoom-in (below) shows a γH2Av focus colocalizing with Rad51, outside a php-mCherry domain. Scale bar = 1 μm. Dotted line outlines nucleus.

**C** Quantification of the number of Rad51 foci per cell in the absence (control, yellow dsRNA, black line) and presence of dUtx dsRNA (orange line). Average number of Rad51 foci per cell is shown at indicated time points. Shade represents SEM of *n* = 4 (control) or *n* = 3 (dUtx dsRNA) biological replicates (with ≥10 cells per condition). *p*-value: 60 min = 0.0067. **D** Quantification of the localization of Rad51 60 minutes after 5 Gy IR either 'inside' a polycomb body or 'outside' a polycomb body. Dots represent *n* = 4 (control) or *n* = 3 (dUtx dsRNA) biological per condition (with ≥10 cells per condition). Straight line in the violin plot indicates the median and dotted lines indicate quartiles. *p*-value: outside = 0.0110. (ns) not significant, (*) p-value ≤ 0.05, (**) p-value ≤ 0.01, (***) p-value ≤ 0.001, two-sided unpaired t-test (**A**, **C**, **D**). Source data are provided as Source Data file.

these results suggest that DSB repair regulation is defective at all heterochromatic sites in the absence of dUtx. We did not find any changes in MMEJ repair pathway choice upon dUtx depletion, suggesting that dUtx, and DSB movement, is mainly important for HR repair (Supplementary Fig. 7E). Finally, loss of dUtx did not affect DSB repair pathway choice at a DR-*white* locus integrated within constitutive heterochromatin, revealing that dUtx only affects DSB repair within facultative heterochromatin (Supplementary Fig. 7F). Importantly, the identified changes in DSB repair pathway choice are not due to indirect cell-cycle effects, since dUtx depletion did not significantly affect cell cycle progression in the Fly-FUCCI system[54] (Supplementary Fig. 7G, H).

To determine whether late HR steps, such as Rad51 binding, important for homology search, can occur within facultative heterochromatin and are dependent on dUtx, we assessed the Rad51 localization pattern at IR-induced DSBs (γH2Av foci) in and outside polycomb bodies (Fig. 5B) in fixed cells. As expected, Rad51 positive γH2Av foci started to increase in control nuclei at 10 min after IR and peaked at 60 min (Fig. 5C, control). Interestingly, we almost never observed Rad51 positive DSBs inside polycomb bodies (Fig. 5D). This result suggests that Rad51 does not bind to DSBs inside polycomb

bodies and only binds to DSBs once these moved outside of polycomb bodies. To test this, we prevented DSB movement by depleting dUtx and assessed the accumulation of γH2Av and Rad51 with respect to polycomb bodies. In control cells, γH2Av levels increased within polycomb bodies 10 min after IR followed by a decrease at later timepoints, indicative of DSB movement (Supplementary Fig. 8A, B). As expected, loss of dUtx resulted in the accumulation of γH2Av foci within polycomb bodies and a reduction outside polycomb chromatin (Supplementary Fig. 8B, C). More importantly, loss of dUtx also resulted in the reduced accumulation of Rad51 foci at DSBs outside polycomb bodies, indicating that Rad51 only binds to the end-resected DSBs once these have moved outside of polycomb bodies (Fig. 5C, D). This observed reduction in Rad51 positive DSBs upon dUtx depletion is in line with the reduced HR levels we identified at facultative heterochromatic DSBs upon dUtx loss (Fig. 5A). Altogether, our results reveal that early HR steps in facultative heterochromatin, such as end-resection, can be performed in the absence of dUtx (Fig. 4E), and that dUtx is specifically required for DSB movement and completion of HR repair (Fig. 5A, C, D).

Finally, to assess the physiological role of dUtx in DSB repair in facultative heterochromatin, we wished to determine whether

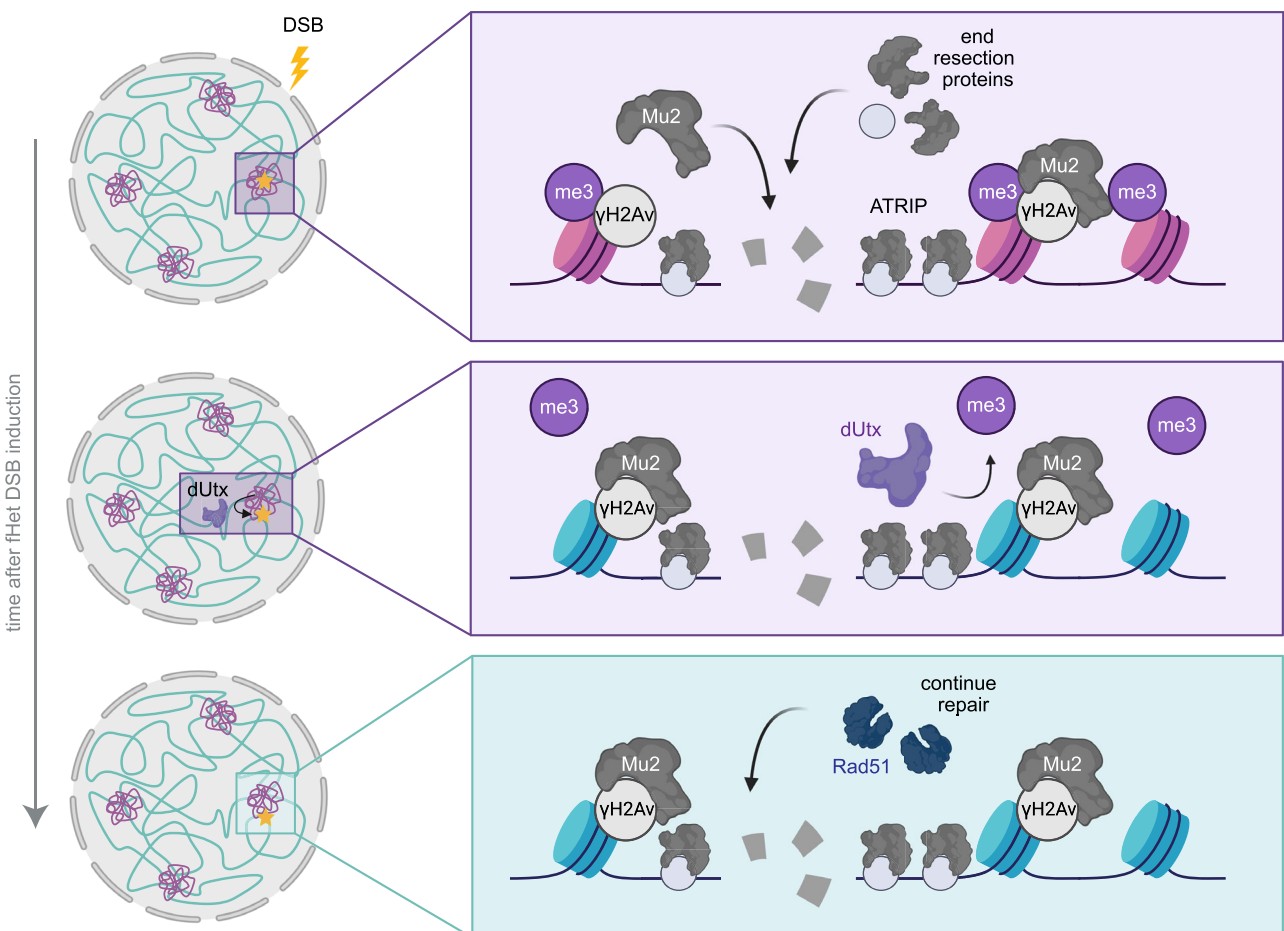

**Fig. 6 | Model for facultative heterochromatin DSB repair.** Facultative heterochromatin (purple) and euchromatin (blue) form distinct domains within the nucleus. Facultative heterochromatin is characterized by H3K27me3 and accumulates in polycomb bodies. DSBs in facultative heterochromatin undergo early steps of HR (i.e. Mu2-recruitment, ATRIP recruitment, end resection) within polycomb bodies. dUtx is required to demethylate H3K27me3 at the DSB site and promote the subsequent DSB movement to efficiently recruit Rad51 and resolve the DSB using HR. Created in BioRender. Janssen, A. (2023) BioRender.com/v65i477.

development of flies mutant for dUtx depend on the presence of intact DNA damage signaling. To do so, we crossed heterozygous dUtx mutant flies with flies that contain a truncation mutation in Ataxia telangiectasia and Rad3 related (ATR, *mei41*). ATR is one of the earliest kinases that rapidly respond to DSB events[55]. Combining dUtx mutant flies with an ATR mutation reduces relative viability by 16%, indeed suggesting that dUtx mutant flies depend on correct DNA damage repair signaling for their development (Supplementary Fig. 8D). More importantly, exposing these mutants to 10 Gy γ-IR during larval development, exacerbated the synthetic lethality observed between ATR and dUtx mutant flies, with 94% relative reduced viability (Supplementary Fig. 8D). These results reveal the physiological role of dUtx and suggests that in the absence of proper facultative heterochromatic DSB repair, flies depend on active DNA damage checkpoint signaling to maintain viability.

## Discussion

Chromatin forms dynamic domains in the nucleus, each characterized by specific molecular properties, which can directly influence the DSB response. However, how transcriptionally inactive facultative heterochromatin (i.e. polycomb chromatin) influences DSB repair remains poorly understood. To address this question and understand how eukaryotic cells maintain the integrity of silenced developmental genes, we here integrated inducible single DSB systems in euchromatin and facultative heterochromatin in *Drosophila melanogaster*. This allowed us to comprehensively study DSB repair in facultative

heterochromatin in animal tissue. We find that DSBs in facultative heterochromatin rapidly move outside polycomb bodies within minutes after their appearance. This movement depends on the H3K27me3 demethylase dUtx. In line with this, we find evidence for dUtx-mediated loss of the silencing mark H3K27me3 near DSBs in facultative heterochromatin. Our data further reveal that early steps of HR (i.e. end resection) can occur efficiently within polycomb bodies and are independent of dUtx, whereas dUtx is required to promote subsequent DSB movement and the completion of HR. Together, we propose a model in which resected DSBs in polycomb bodies are subjected to dUtx-mediated loss of the silencing mark H3K27me3, which in turn promotes DSB movement and timely repair by homologous recombination (Fig. 6).

Specific movements of DSBs have been identified to occur in a variety of chromatin compartments including centromeres[16], nucleoli[17–19] and pericentromeric constitutive heterochromatin[16,20–22]. Movement of these DSBs has been suggested to promote binding of HR proteins[17,18], as well as prevent aberrant recombination between repetitive sequences[20,21]. Here, we find that DSBs move outside polycomb bodies in vivo (Fig. 2). In contrast to centromeres, nucleoli and constitutive heterochromatin, facultative heterochromatin mainly contains unique sequences and is deprived of repetitive sequences. We therefore propose that DSB movement in facultative heterochromatin did not evolve to prevent aberrant recombination, but rather reflects the necessity to create a repair-competent state, facilitating access to the DSB repair machinery (Fig. 6).

Our data suggest that the movement of DSBs outside polycomb bodies is directly regulated by the dUtx-mediated removal of H3K27me3 at the break site (Figs. 3 and 4). H3K27me3 is required to recruit PcG proteins to enhance compaction and maintain a silenced state[9–13]. Therefore, active removal of H3K27me3 at the DSB site by dUtx can also directly lead to a local loss of PcG proteins (Supplementary Fig. 5E). This can subsequently lead to changes in the molecular- and biophysical- properties of the DSB locus, creating an environment distinct from the surrounding polycomb body and the active expulsion or passive separation of the DSB from the polycomb body. In line with this hypothesis, we find that in the absence of dUtx, and the subsequent retention of H3K27me3, DSBs remain longer within the polycomb body (Fig. 4). These results reveal analogies with our previous findings at DSBs in constitutive heterochromatin, where we found that loss of the silencing mark H3K9me3 at DSBs by the histone demethylase dKDM4A ensures DSB movement outside the constitutive heterochromatin domain[32,33].

Since we did not observe a complete inhibition of facultative heterochromatic DSB movement upon dUtx depletion, additional processes are likely involved, such as DSB end processing or the recruitment of specific chromatin- or repair- proteins. End resection as well as chromatin proteins, including the cohesin- and SMC5/6- complexes, drive DSB movement in other chromatin domains[16,18,20], suggesting that additional components could be driving movements in polycomb bodies.

Our findings indicate that dUtx-mediated demethylation of H3K27me3 at facultative heterochromatic DSBs is important for repair pathway choice, since dUtx loss shifts the choice towards NHEJ, resulting in decreased HR (Fig. 5). Considering that dUtx depletion only affects DSB repair pathway choice in facultative heterochromatin, not euchromatin, these repair pathway changes are unlikely to be driven by indirect general defects in cell cycle progression (Supplementary Fig. 7G, H) or transcriptional regulation of repair genes[56]. Therefore, we hypothesize that the HR/NHEJ repair pathway choice at facultative heterochromatin could be directly regulated by dUtx through two non-mutually exclusive mechanisms: (1) promoting DSB movement, and (2) direct impact on binding of HR- or NHEJ- proteins at DSB sites.

In the first model, dUtx is promoting HR by moving the DSB to a more HR-prone chromatin state depleted of silencing marks. The movement might therefore specifically facilitate the access to 'late' HR repair proteins (e.g. Rad51 loading (Fig. 5D) or helicases to resolve D-loops), usually excluded from the compact polycomb state. Moreover, moving an HR-proficient DSB away from the compact facultative heterochromatin might provide the required chromatin mobility necessary to perform homology search[57]. Indeed, we observe that loss of dUtx has no impact on the initial stages of HR, such as end resection and ATRIP loading, in polycomb bodies. However, dUtx loss does impede DSB movement, and subsequent later HR steps as evidenced by the decreased number of HR repair products identified at facultative heterochromatic DSB sites (Fig. 5). Interestingly, we find the majority of DSBs in facultative heterochromatin to be dependent on NHEJ, while we also observe movement of most, if not all, DSBs. This could indicate that most DSBs move, regardless of DSB repair pathway choice. However, the fact that dUtx loss increases repair by NHEJ and delays movement, suggests that NHEJ can occur irrespective of H3K27me3 loss and DSB movement, while HR specifically depends on this DSB movement. Finally, the low number of MMEJ events was not affected by the absence of dUtx (Figure. S7E), indicating that MMEJ repair also occurs independently of DSB movement.

In our second model, we propose that the decreased frequency of HR repair in the absence of dUtx is caused by changes in binding of repair proteins to histone modifications at the break site. It is possible that specific NHEJ proteins directly bind H3K27me3 within the polycomb body. Alternatively, H3K27me1 or unmethylated H3K27

residues, as a result of dUtx-mediated H3K27me3 demethylation, could directly recruit proteins important for HR. In line with this hypothesis, previous work identified that the HR-promoting TONSL-MMS2L complex has a higher affinity for unmodified histones, generated following DNA replication[58].

Although DSBs at the *3fH_1* locus did not induce evident H3K27me3 loss (Fig. 3A) this region showed a clear defect in HR in the absence of dUtx (Fig. 5A). This indicates that dUtx plays an important role in DSB repair at this integration site. The fact that we did not identify loss of H3K27me3 at this site could suggest that H3K27me3 demethylation has a relatively transient nature, potentially counteracted by histone methylation activities. Depending on the location within heterochromatin, these histone methylation activities might differ in activity. Although H3K27me3 is currently the sole target described for dUtx[48,49], we cannot exclude that dUtx plays additional roles in repair, which could be independent of H3K27me3 demethylation and may play a role only at specific heterochromatic loci.

Despite differences in dUtx-dependency for repair pathway choice, we find the frequencies of HR and NHEJ repair pathway usage in facultative heterochromatin and euchromatin to be similar in wild type *Drosophila* tissue (Fig. 1, Supplementary Figs. 1 and 3). These results are consistent with previous findings in which DSBs in H3K27me3-enriched imprinted loci in mice did not differ in repair pathway usage when compared to the corresponding active allele[59]. In addition, both HR and NHEJ components are recruited to laser-damaged inactive X chromosomes in female human cells[25]. In contrast, a recent study that used a sequencing-based reporter system in cancer cells to investigate the impact of chromatin on CRISPR-Cas9-induced DSBs did reveal relative differences in repair pathway usage in H3K27me3-enriched regions[60]. The authors found a relative decrease in NHEJ and concurrently an increase in usage of MMEJ within H3K27me3-enriched regions. In contrast to our results, these findings suggest that end resection-based repair pathways are preferred at facultative heterochromatic DSBs. These different outcomes could indicate differences in repair pathway usage between species or could be due to differences in the approach used to induce DSB induction (CRISPR-Cas9 versus I-SceI). Moreover, heterochromatin properties in vivo may vary from that observed in cultured cells, potentially leading to disparate outcomes.

In conclusion, our work demonstrates that DSBs in facultative heterochromatin require specific local chromatin changes and DSB movements for their faithful repair in animal tissue. Our results emphasize the importance of understanding how different chromatin components influence DSB repair pathway choice and maintain genome stability across diverse chromatin domains. Facultative heterochromatin regions are often associated with high mutational loads in cancer[61], indicating that these domains are particularly vulnerable to aberrant DNA damage repair. Moreover, the human homolog of dUtx, UTX, is often mutated in cancer[62]. In the long-term, research into DNA damage repair in heterochromatin will give insights into how misregulation of chromatin proteins, such as UTX, could result in increased genome instability and specific mutational signatures in cancer, ultimately contributing to disease development.

## Methods
### Constructs
The DR-*white* construct was created previously[22]. For the generation of the mCherry-ph-p plasmid, ph-p was N-terminally tagged with mCherry and cloned into a pCasper5 vector for random p-element transformation in flies. For cell culture experiments, ph-p, H3K27me3-mintbody and ATRIP were cloned into pCopia vectors containing N-terminal mCherry (ph-p), C-terminal mCherry (H3K27me3-mintbody) or C-terminal eYFP (ATRIP) epitope tags. ph-p was cloned from a pFastBac plasmid (Addgene #1925), ATRIP was cloned from cDNA generated from RNA extracted from wild type flies and the

H3K27me3-mintbody was received in a plasmid (pUAST_2E12LI-sfGFP) as a kind gift from Dr. Hiroshi Kimura.

## Fly lines and genetics

Flies were reared at room temperature on standard medium, except otherwise specified. Embryo injection and generation of new DR-*white* and ph-p-mCherry fly lines were performed by BestGene, Inc (Chino Hills, CA, USA). DR-*white* attB containing plasmids were integrated in Minos-mediated integration cassette (MiMIC) integration sites as described previously[22]. An overview of the MiMIC integration sites to generate DR-*white* fly lines can be found in Supplementary Table 1. Facultative heterochromatin integration sites were selected at non-gene coding regions with high H3K27me3 levels in OregonR flies (modEncode), as well as near a gene known to be regulated by H3K27me3 and/or PcG proteins[63]. Euchromatin integration sites were selected based on low H3K27me3 and low H3K9me2/3 levels. To create ph-p-mCherry fly lines, pCasper5-ph-p-mCherry plasmid containing the copia promoter and P-element transposons were injected in embryos of w1118 flies. To induce knockdown of either DmCtIP, DmKu70, DmRad51 or dUtx, flies containing Gal4 driven by an Actin5C or daughterless promoter were crossed with *UAS- DmCtIP/ DmKu70/DmRad51/dUtx dsRNA* flies. A list of all fly lines used can be found in Supplementary Table 1.

Synthetic lethality without irradiation (Supplementary Fig. 8D, left) was assessed by calculating the percentage viability of dUtx mutants using the ratio of adult males heterozygous for *dUtx [f01321]* compared to wild type offspring. The percentage viability of ATR and dUtx double mutants was calculated using the ratio of adult males hemizygous for *mei41[29D]* and heterozygous for *dUtx [f01321]* compared to males hemizygous for *mei41[29D]* and wild type for *dUtx*. *Mei41[29D]* resides on the X chromosome and hemizygous mutant males without exogenous DNA damage do not have significant viability problems[64]. For synthetic lethality experiments in the presence of irradiation (Supplementary Fig. 8D, right) all fly lines were irradiated with 10 Gy γ-irradiation using an IBL 437 C machine with a cesium-137 gamma source at day 4 following egg laying. The percentage viability of dUtx mutants 1–3 days after eclosion was calculated using the ratio of adult males heterozygous for *dUtx [f01321]* compared to wild type offspring. The percentage viability of ATR mutants was calculated using the ratio of adult *mei41[29D]* mutant hemizygous males compared to wild type males. The percentage viability of ATR and dUtx double mutants was calculated using the ratio of adult males hemizygous for *mei41[29D]* and heterozygous voor *dUtx [f01321]* compared to males wildtype for ATR and dUtx. All genotypes analyzed were quantified from five to 15 crosses.

## ChIP-qPCR

Third instar DR-*white* larvae ( + and - *hsp.I-SceI*) were heat-shocked for 1 h to express I-SceI. Six hours after I-SceI activation, 40 larvae per condition were pooled together to extract chromatin following the ChIP protocol of the Kevin White lab (https://www.encodeproject.org/documents/f890fde6-924c-4265-a60f-c5810401066d/), with slight adjustments. In short, larvae were homogenized in buffer A1 (60 mM KCl, 15 mM NaCl, 15 mM HEPES pH7.6, 4 mM MgCl₂, 0.5% TritonX-100, 0.5 mM DTT, protease inhibitor (Roche #1873580)) and fixed using 1.8% paraformaldehyde (EMS). Fixation was stopped by addition of glycine. After several washing steps with buffer A1, nuclei were isolated (140 mM NaCl, 15 mM HEPES pH7.6, 1 mM EDTA, 0.5 mM EGTA, 0.1% sodium deoxycholate, 1% TritonX-100, 500 µM DTT, protease inhibitors, 0.1% SDS and 0.5% N-lauroylsarcosine) and the chromatin was fragmented by sonication for 10 cycles on Bioruptor (Diagenode high settings, 30″ on/off). Chromatin was separated from cell debris using centrifugation and stored at −80 °C for a maximum of 2 months.

ChIP was performed as described earlier[65] using 1–2 µg chromatin and 1–5 µg antibody. ChIP antibodies used were mouse anti-γH2Av (Developmental Studies Hybridoma Bank, UNC93-5.2.1), rabbit anti-H3K27me3 (Invitrogen, MA5-11198), rabbit anti-H3 (Abcam, ab1791), H2AK119Ub (Cell Signaling, 8240S) and H3K9me3 (Abcam, 176916). Enrichment of specific histone marks at the DR-*white* locus was quantified by qPCR using SYBR Green Master Mix (Roche) and qPCR primers 1.4 kilobases and 3.1 kilobases away from the DSB (3xp3) as well as primers for an internal control. H3K27me3 levels at 3xp3 were normalized to H3K27me3 levels at the *ubx* gene, since it has consistently high H3K27me3 levels[63]. qPCR analyses were performed using Bio-Rad CFX96 Touch Real-Time PCR qPCR system.

Epicypher SNAP-ChIP K-MetStat panel (19-1001) was used to validate the specificity of the H3K27me3 antibody used for ChIP (Supplementary Fig. 1A). The SNAP-ChIP panel contains barcoded nucleosomes with a specific methylation mark (unmethylated H3, H3K4me1/me2/me3, H3K9me1/me2/me3, H3K27me1/me2/me3, and H3K36me1/me2/me3). A total of 0.4 µL of 0.6 nM K-MetStat stock per 2 µg of chromatin was added at the start of the ChIP procedure. Subsequent qPCR analysis using primers specific to unique barcodes corresponding to each modified nucleosome in the panel allows the quantification of antibody specificity and efficiency.

## DR-white repair product analysis

Quantification of somatic repair products in DR-*white*, I-SceI larvae was performed, as described previously[22]. In short, either I-SceI was expressed using heat-shock in third instar DR-*white*/hsp.I-SceI larvae, 24 h before single larvae were collected. Alternatively, ecDHFR-I-SceI was stabilized by feeding DR-*white*/ecDHFR-I-SceI 80 µM trimethoprim (Sigma) throughout development (3–4 days). To prepare trimethoprim containing food, 1.67 g instant *Drosophila* medium (Formula 4–24, Carolina Biological Supply) was mixed with 5 mL non-distilled water containing 5.3 µL of 100 mM trimethoprim while vortexing.

To analyze repair products, the upstream *white* gene was PCR amplified and the PCR product was treated with ExoSAP-IT to enzymatically remove excess primers and unincorporated nucleotides, followed by Sanger sequencing. Analysis of Sanger sequences was performed using the TIDE (tracking of indels by decomposition) algorithm[37]. HR repair products were identified by loss of the I-SceI cleavage site and appearance of the wildtype *white* gene, which is essentially a 23-nucleotide deletion at the I-SceI cut site. NHEJ products were identified as insertions and deletions up to 25 bp, except for the 23-nucleotide deletion. For PCR and sequencing primers, see Supplementary Table 2.

To analyze repair products using next-generation sequencing (NGS), the upstream *white* gene was PCR amplified using our standard DR-*white* PCR primers and purified using AMPure XP beads (Beckman Coulter) according to the manufacturer's protocol and DNA was eluted in 20 µl MQ. An additional nested PCR was performed using primers containing the p5 and p7 index sequences. The PCR products were purified with AMPure XP beads and eluted in 20 µl MQ. PCR samples were pooled at equimolar concentrations per target-specific PCR and sequenced on an Illumina NextSeq2000 by 150-bp paired-end sequencing. For PCR primers, see Supplementary Table 2.

To analyze repair outcomes from NGS of PCR products SIQ v1.3[66] was used. In short, paired-end sequence data files were selected using the graphical user interface. A reference FASTA file was provided containing the DR-*white* sequence including the I-SceI site. For the target site the following sequences were set in SIQ: left flank TTGAGCTGTAGGGATAA, right flank CAGGGTAATAGCTCTTTG. The following primer sequences were used: left primer GACTGGACT CATTTACCGCCC, right primer TTGGTAGGACACTGGGCAC. The repaired DR-*white* construct without the I-SceI cut site was used as HR sequence and SIQ was run with standard settings. The resulting output of SIQ was further processed with SIQPlotteR using the following

setting: filter reads by distance from expected cut site to ≤10 bp to exclude events that do not occur at or near the I-SceI target site. The repair products identified as 1 bp insertion, deletion, deletion with insert, deletion with templated insert and insertions, were divided into NHEJ and MMEJ based on the length of the microhomology that these products contained. MMEJ is defined as deletion products containing microhomologies of 2 to >4 bp, whereas the remaining insertion and deletion repair products are defined as NHEJ.

## Immunofluorescence staining

For immunofluorescence (IF) staining on wing discs, wing discs were dissected from third instar DR-*white* larvae and fixed with 4% PFA and 0.1% Triton-X on slides for 15 min, as described earlier[67]. Slides were dipped in liquid nitrogen and stored in 96% ethanol at −20 °C until staining procedure. Before staining, slides were thawed at room temperature and washed in PBS for 20 min.

For IF staining on cells in culture, cells were fixed by incubating cells with 4% PFA and 0.1% Triton-X for 15 min. After washing, the cells were further permeabilized by incubating them with 0.4% Triton-X diluted in PBS for 20 min until staining procedure.

To continue IF on wing discs and cells, imaging slides were blocked using 0.4% Triton in PBS and 5% milk for 1 h at room temperature. Primary antibody incubations were performed overnight at 4 °C in block buffer. Primary antibodies used were mouse anti-γH2Av (1:250, Developmental Studies Hybridoma Bank, UNC93-5.2.1), chicken anti-mCherry (1:1000, abcam, ab205402) and rabbit anti-Rad51 (1:500, kind gift from Dr. Irene Chiolo). Slides were washed 3 times with block buffer. Secondary antibody incubation was performed at room temperature in PBS 0.4% Triton for 2 h. Secondary antibodies used were Alexa 488 goat anti-mouse (1:600; Invitrogen, A11029), Alexa 568 goat anti-chicken (1:600; Invitrogen, A11041) and Alexa 647 goat anti-rabbit (1:600; Invitrogen, A21245). Slides were subsequently washed 3 times with 0.4% Triton in PBS, incubated with 3 μg/ml DAPI for 30 min, washed with PBS and mounted using Prolong Diamond Antifade Mountant and a 20 × 20 mm #1.5 coverslip.

## Cell culture

Kc cells (DGRC) were cultured in CCM3 medium (Avantor) supplemented with Antibiotic Antimycotic Solution (Sigma) at 27 °C. S2 cells (DGRC) were cultured in Schneider's Insect medium (Sigma-Aldrich, S0146), supplemented with Penicillin-Streptomycin and FBS at 27 °C. Kc and S2 cell lines were authenticated by DGRC upon purchase. To express ph-p-mCherry and ATRIP-eYFP, Kc cells were transiently transfected with 400 ng of each plasmid using the TransIT-2020 reagent (Mirus). To express H3K27me3-mintbody-mCherry, S2 cells were transiently transfected with 300 ng plasmid using the TransIT-2020 reagent. Live imaging was performed 72 h after transfection. To induce RNAi-mediated depletion of dUtx, cells were transfected with 5–10 ug dsRNA (TransIT-2020 reagent (Mirus)) and harvested or subjected to imaging 3 days later. dsRNA was generated using a MEGA-Script T7 transcription kit (Life Technologies). PCR products containing a T7 promoter sequence and the target regions were used as templates (Supplementary Table 2). Irradiation was performed by exposing cells to 5 Gy of γ-rays in an IBL 437 C machine with a 137-Cs source.

## RT-qPCR

RNA was isolated by homogenizing either single larvae or Kc cells in 200 μL Trizol (Invitrogen) using an electrical douncer (VWR). After addition of 40 μL of chloroform and centrifugation, RNA from the aqueous phase was precipitated using isopropanol and further purified using an ethanol wash step. cDNA was synthesized using iScript following standard cDNA synthesis protocol (Bio-Rad). qPCR was subsequently performed on cDNA with gene-specific primers (Supplementary Table 2).

## Imaging

Images of fixed tissue and cells, as well as live Fly-FUCCI wing discs, were acquired using a 60× oil immersion objective (NA 1.42) on a DeltaVision microscope (DeltaVision Spectris; Applied Precision, LLC). For all DSB tracking experiments in wing discs as well as H3K27me3-mintbody signal analyses at DSBs in cells, time-lapse images were acquired using a LD C-Apochromat 63×/1.15 W Korr M27 objective on a LSM880 microscope with Airyscan (Zeiss), and images were processed using the Zeiss ZEN software. Time-lapse images were acquired once every 5–10 min. Image analysis and focus tracking were performed manually using the Fiji image analysis software.

For live imaging of wing discs, third instar larvae were dissected and wing discs were placed on a slide in 10 μL of Schneider S1 medium supplemented with 10% FBS and covered with a 20 × 20 mm #1.5 coverslip, as described previously[68]. To stabilize ecDHFR-I-SceI in larval tissue, 400 μM trimethoprim (Sigma) was added 15 min prior to imaging.

## Reporting summary

Further information on research design is available in the Nature Portfolio Reporting Summary linked to this article.

## Data availability

The raw targeted sequencing data generated for this study have been been deposited to BioProject accession number PRJNA1132713. in the NCBI BioProject database (https://www.ncbi.nlm.nih.gov/bioproject/). Source data are provided with this paper.

## Code availability

The SIQ software to analyze repair junctions has been published[66]. The online tool to analyze repair junctions is available at: https://siq.researchlumc.nl/SIQPlotter/.

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

## Acknowledgements

We thank Jurian Schuijers and Lucie van Leeuwen for critically reading our manuscript and all the members of the Lens and Janssen laboratories for their valuable input during lab meetings. Special thanks to Johannes Lehmann for his advice on live imaging analyses and Robin Geene for his advice on Illumina sequencing. This work was funded by the European Research Council (ERC) under the European Union's Horizon 2020 research and innovation program, grant agreement No. 850405, and VIDI VI.Vidi.203.001 financed by the Dutch Research Council (NWO). Several figures were created with BioRender.com.

## Author contributions

M.R.W. performed most experiments and analysis. A.D. assisted with experiments in Fig. 3 and Supplementary Fig. 2. R.v.S. and M.T. supervised Illumina sequencing experiments and performed bioinformatic analysis in Supplementary Fig. 3 and Supplementary Fig. 7E. A.K. and J.L. assisted with *Drosophila* Melanogaster culture and cloning. A.J. and S.U.C. performed experiments in Supplementary Fig. 8D. A.J. and M.R.W. contributed to project planning, experimental design, interpretation of results, and manuscript preparation.

## Competing interests

The authors declare no competing interests.
