## [Peer Review File · Nature Communications]

Double-strand breaks in facultative heterochromatin require specific movements and chromatin changes for efficient repairREVIEWER COMMENTS

Reviewer #1 (Remarks to the Author):

In this study, Wensveen et al. investigated the dynamic response to double-strand breaks (DSBs) within facultative heterochromatin *in vivo* by integrating an inducible single DSB system into fruit flies. Facultative heterochromatin regions are found in nuclear structures called polycomb bodies and are enriched for H3K27me3. Using live-cell imaging, they show that DSB-signaling and repair occur with similar kinetics in euchromatin and facultative heterochromatin regions. In contrast, DSB repair that occurs in facultative heterochromatin is only observed after the migration of the break outside of the polycomb bodies, a phenomenon that is reminiscent of DNA repair that has been characterized in constitutive heterochromatin. Like the previously reported role of H3K9me3 demethylase dKDM4A in DSBs' movement and repair that occur in constitutive heterochromatin (Janssen et al. *G&D* 2019), depletion of the H3K27me3 demethylase dUTX reduced both DSBs movement and repair in facultative heterochromatin. Strikingly, while dKDM4 drives DNA repair pathway choice in favor of NHEJ, dUTX favors DNA repair by HR in facultative heterochromatin.

In a nutshell, the authors adapted a well-established assay to enable the characterization of the spatiotemporal dynamic of DNA repair in facultative chromatin, revealing that DSBs occurring in facultative and constitutive heterochromatin are resolved by distinct DNA repair pathways. While the proposed model is attractive and is of interest to a broad audience, some points need addressing before this study can be considered for publication in *Nature Communication*.

Major:

Data supporting the role of dUTX in promoting DNA repair pathway choice in an H3K27me3-dependent manner should be strengthened. In the current version of the manuscript, the link between H3K27me3 levels – dUTX and DNA repair by HR is only supported by observations made at one locus (fHet1) where concomitant depletion of dUTX leads to a reduction in H3K27me3 levels and reduced HR. At the other locus only one of these phenotypes is observed. Can dUTX be knocked out or inhibited through orthogonal approaches to define if a stronger depletion/inhibition of the demethylase improves H3K27me3 depletion at all the locus? In addition, the authors should show that the effect is specific to facultative heterochromatin by quantifying the impact of dUTX depletion on DSB repair at the locus of constitutive heterochromatin. Are these phenotypes all recapitulated in IR-treated samples?

Decompaction of the chromatin has been linked with an increased level of RAD51 at the locus of DSB and increased HR in constitutive heterochromatin (Chiolo et al *Cell* 2011). In contrast, depletion of dUTX results in less HR repair in facultative heterochromatin. According to these observations, depletion of dUTX shouldn't promote the recruitment of RAD51 to polycomb bodies. Validating this hypothesis would further support their model. This can be achieved by investigating if dUTX impacts chromatin compaction and/or the recruitment of RAD51 in polycomb bodies. Further characterizing DNA repair events that occur inside and outside of the polycomb bodies

(RPA, RAD51) is essential here given the different phenotypes associated with histone H3 demethylation in facultative and constitutive heterochromatin.

Reviewer #2 (Remarks to the Author):

In this manuscript, Wensveen and colleagues contribute to the ongoing body of work to determine double-strand break repair dynamics and factors that influence repair and repair pathway choice in multicellular organisms. Specifically, they investigate the factors that dictate DSB repair in facultative heterochromatin. They utilize *in vivo* systems to induce DSBs in both euchromatin control and facultative heterochromatin domains and determine the frequencies of DSB induction, rates of DSB localization within the Polychrome regions, and repair pathway choice. While overall DSB induction and proportion of HR vs. NHEJ with indels did not differ overall between euchromatin and facultative heterochromatin, they found that repair by HR was facilitated but Utx-mediated demethylation of H3K27me₃, leading to re-localization of the break outside of the polycomb bodies to facilitate completion of HR.

This is a well-developed study, including the establishment of new assays and tools to analyze DSB formation, localization and repair within facultative heterochromatin. Strong methods with sound controls effectively leading to a clear model depicting movement of DSBs in these regions of the genome. It will have an impact in the field regarding the complexity of the mechanisms that ensure DSB repair in various contexts within the genome. I have no major issues to address with this work.

A few minor suggestions:

Experiments/Results:

Shifting between the two I-SceI induction systems seems warranted and justified. However, the ecDHFR system is more or less constitutively active throughout all of development (at least during larval stages), whereas the heat shock system is induced at third instar larvae. Is there any data to suggest that DSB repair kinetics and factors may be different in 3rd instar larvae than breaks that are induced as early as 1st instar larvae? I do understand it may be experimentally impossible to induce the ecDHFR system in a more defined time point like the heat shock, but the authors may want to comment on this difference between the two systems.

It is interesting that a fair amount of DSBs are unresolved within the 60 minute window that it is observed (~16-50% depending on the induction method and what was being analyzed; almost 100% in IR-induced – S2C)). Do you know the kinetics to repair all events (>60 minutes)? Related, do you have a non-heat shock or non-TMP treated controls to determine whether some of the DSBs that are being analyzed are persisting from earlier points in development due to leaky expression of the I-SceI transgene?

I also find it interesting that endogenous levels of Utx are important for survival in ATR-mutant

background (S4H). Considering the deficiencies in repairing induced DSBs with siRNA KD of Utx, is there an increase in sensitivity to IR in the double mutants? dATR single mutants are hypersensitive to IR to begin with, so it may require lower doses to detect differences in the double mutants.

Methods/Statistics

Throughout the description of the methods and results (i.e., Fig. 1A), the authors refer to NHEJ with insertions and deletions (indels) as “NHEJ”. It is important to note that this system only detects NHEJ with indels (precise NHEJ is nicely noted in the figure). Related, “HR” should be defined some point as intrachromosomal HR (to contrast to inter-sister HR; line 574). This is helpful for those outside of the field that don’t fully understand these nuances.

Fig. S1D, the authors state that the relative proportion of HR vs. NHEJ is not different between lines. Where statistics completed on this?

Fig. 2F, which fHet line was used?

Fig. 3B, values are “ns”, but relative to what?

Fig. S4A, the “#” is defined as DSBs that were not resolved in the figure legend. However, the colors of the bars suggest some of them were (with white = resolved). For example, S4A, bottom two events suggest it was resolved based on white coloring at 20 minutes. But the “#” suggests they were not resolved. Is this because imaging was cut short for these events? If yes, maybe the white coloring that indicates “resolved” should be changed to “n.d.” (gray)? It’s also unclear why this figure, and none of the others, needed to highlight the unresolved DSBs with “#”.

It wasn’t clear at first that each of the euchromatin sites and fHet sites were different lines. Adding this in the Fig. 1 legend or early into the manuscript (lines 87-90) would be helpful.

How did you irradiate the wing discs? X-ray, gamma? Please include in methods.

Minor typos and citations:

Gene names should be italicized, although this may be journal-specific.

Line 92, “NHEJ will generate” should be changed to “NHEJ may generate”.

Line 241, *Melanogaster* should be *melanogaster*

High levels of H3K27me3 in Ubx is stated in several figure legends. This will be helpful to cite, unless it is common knowledge. Perhaps adding a citation in the methods on how the relative H2K27me3 levels were calculated?

Reviewer #3 (Remarks to the Author):

The paper by Wensveen et al. entitled “Double-strand breaks in facultative heterochromatin require specific movements and chromatin changes for efficient repair” describes a DR-white reporter for the in vivo analysis of DSB repair in two euchromatin and three heterochromatin domains in *Drosophila*. Using this reporter, the work suggests that loss of the chromatin mark H3K27me3 by dUtx is required for DSB movement from polycomb bodies and subsequent repair by homologous recombination in heterochromatin, but not euchromatin. The use of these reporters in an animal model is very elegant and provides insight into DSB repair in different chromatin compartments at the organismal level. However, although the work in its current form is interesting, the data are rather preliminary and do not warrant strong conclusions with regard to DSB regulation in heterochromatin versus euchromatin, limiting the quality and novelty of the study. The work also lacks the depth required for publication, as it leaves numerous questions unanswered.

Major concerns:

- The majority of repair in the DR-white reporter occurs via NHEJ (Figure 1I). How can this be explained since in this type of DR reporters (similar to that in the well-established DR-GFP reporter for homologous recombination) are mostly used to study homologous recombination? Which NHEJ pathways act on DSBs in this reporter? This could be deduced from repair junction analysis and should be further validated by *Drosophila* Ku and PolQ knockdown for cNHEJ and MMEJ.
- DSBs move away from heterochromatin for repair, which is dependent on dUtx and may be required for repair by homologous recombination. However, the majority of repair in the DR-white reporter occurs via NHEJ (see also previous point). Does NHEJ repair in heterochromatin also require DSB movement? This is particularly important given that the Van Steensel lab reported that MMEJ predominantly occurs in this chromatin context (Schep et al., Mol Cell, 2021). A more extensive analysis of DSB movement and repair pathway choice would increase the novelty of the work.
- The authors have studied only one heterochromatin mark on one side of the induced DSBs, and base all their conclusions with regard to DSB movement and repair on this analysis. Additional heterochromatin marks should be studied and the analysis of these marks should be extended to a larger regions flanking either side of the induced DSBs.
- Two out of three DR-white reporters in heterochromatin show an H3K27me3 reduction, one of which was studied and showed movement of DSBs for repair. This is too limited to draw firm conclusions about the necessity of movement for DSB repair in heterochromatin. Does the second heterochromatin DSB showing no reduction in H3K27me3 also move? If so, then movement is not dependent on loss of H3K27me3.
- RNAi against dUtx reduced homologous recombination in the DR-white at all three heterochromatic loci, while loss of H3K27me3 was only affected at two of these loci. This suggests that repair at these loci is not fully dependent on H3K27me3 loss, which would not be in line with

the main conclusion of the work. Moreover, it also suggests that dUtx has roles in repair beyond its function in removing H3K27me3, yet how remains unclear. Does Utx modify other targets involved in repair?

- The dominant type of repair in the DR-white reporter is NHEJ. Does RNAi against dUtx also impact this repair pathway and are there specific effects in euchromatin versus heterochromatin?

- How specific is the single RNAi against dUtx? RNAi is known for its off-target effects, so a multiple RNAi approach or RNAi complementation approach should be considered to address this (particularly with regards to the two previous points).

- H3K27me3 loss and DSB movement are required for homologous recombination, which depends on CtIP-dependent end-resection. But are H3K27me3 loss and/or DSB movement dependent on CtIP-dependent end-resection (or in other words are these processes affected by CtIP depletion)? If not, what does this mean for repair by HR versus NHEJ?

Minor concerns:

- The DR-white reporter resembles known reporters, most notably DR-GFP (initially developed by Maria Jasin's lab and used in the field for decades). Although extensive validation may not be needed, using CtIP knockdown for validation of homologous recombination of DSBs in the reporter is fairly limited. Also, CtIP is not uniquely involved in homologous recombination and also plays a role in MMEJ. Knockdown of core homologous recombination factors such as BRCA1, BRCA2, PALB2 or RAD51 would be required for further validation.

We appreciate the time the reviewers took to critically assess our manuscript and we thank them for their insightful comments. We believe our manuscript has significantly improved by addressing their feedback. Below is our point-by-point response to all comments.

Of note; in our revised version, we have changed the names of the DR-white lines, which we will refer to in answering the reviewers' comments. This to be more consistent with DR-white annotations used in Janssen et al., Genes & Dev. 2016 and Janssen & Colmenares et al., Genes & Dev. 2019.

Reviewer 1.

1. Data supporting the role of dUTX in promoting DNA repair pathway choice in an H3K27me3-dependent manner should be strengthened. In the current version of the manuscript, the link between H3K27me3 levels – dUTX and DNA repair by HR is only supported by observations made at one locus (fHet1) where concomitant depletion of dUTX leads to a reduction in H3K27me3 levels and reduced HR. At the other locus only one of these phenotypes is observed. Can dUTX be knocked out or inhibited through orthogonal approaches to define if a stronger depletion/inhibition of the demethylase improves H3K27me3 depletion at the all the locus?

We thank the reviewer for their comment. We currently do not have a way to improve dUtx depletion or inhibition in our single DSB systems *in vivo* and therefore focused our efforts on performing approaches to strengthen the link between H3K27me3 levels – dUtx and HR repair in cells in culture:

- To strengthen the link between H3K27me3 levels and dUtx, we have now included an orthogonal assay to determine H3K27me3 levels at DSBs in polycomb bodies in addition to our H3K27me3 ChIP data (**Fig.3A,D**). In short, we employed an H3K27me3 mint-body, which is a fluorescently tagged H3K27me3 antibody developed by the lab of Hiroshi Kimura (<https://pubmed.ncbi.nlm.nih.gov/21576221/>) (**Fig.3E-G**). This mint-body allows for the live imaging analysis of changes in H3K27me3 levels. Using this approach, we find a ~10% reduction in H3K27me3 levels at DSBs in polycomb bodies (following 5Gy IR), already before these DSBs move outside the heterochromatin domain (**Fig.3G**). This reduction in H3K27me3 levels at DSBs is completely prevented upon dUtx depletion, reinforcing our conclusion that dUtx is responsible for the observed loss of H3K27me3 at heterochromatic DSBs.

- To further strengthen the link between dUtx and HR repair in facultative heterochromatin, we have now assessed the localization pattern of the HR protein Rad51 using immunofluorescence following irradiation of cultured cells (**Fig.5B-D**). We find that in control cells, Rad51 solely localizes outside polycomb bodies and we almost never observe Rad51 at DSBs inside polycomb bodies. When we deplete dUtx, we prevent movement of DSBs outside polycomb bodies (**Fig.4C**) and observe a significant reduction in Rad51 positive DSBs outside polycomb bodies (**Fig.5D**). In addition with the identified reduction in HR repair products (**Fig.5A, Fig.S7B-D**), this reinforces our conclusion that dUtx-mediated DSB movement is important to promote later HR steps (e.g. Rad51 loading).

Together, these results strengthen our proposed link between dUtx-mediated H3K27me3 loss, DSB movement and HR repair in facultative heterochromatin.

2. In addition, the authors should show that the effect is specific to facultative heterochromatin by quantifying the impact of dUTX depletion on DSB repair at the locus of constitutive heterochromatin.

We agree this is an important point and have included this experiment in the revised manuscript. We do not find an effect of dUtx depletion on DSB repair pathway choice at a DR-*white* locus in constitutive heterochromatin (**Fig.S7F**).

3. Are these phenotypes all recapitulated in IR-treated samples?

We have reproduced all phenotypes using IR-treated samples:

- Loss of H3K27me3 levels are reduced at IR-induced DSBs as assessed by our H3K27me3 mint-body analyses (**Fig.3E-G**). This H3K27me3 loss is prevented upon dUtx depletion.
- Similar to single (I-SceI dependent) DSBs *in vivo*, we find DSB movement upon IR of tissue *in vivo* (**Fig.2F**) and cells in culture (**Fig. 4C, Fig.S6A**). This DSB movement is delayed upon dUtx depletion (**Fig.4C, Fig.S6B**).
- As also discussed above at point 1, we find a reduction in Rad51 positive DSBs (γ H2Av foci) upon dUtx depletion (**Fig. 5B-D**). These DSBs were induced using 5Gy IR of cells in culture.

4. Decompaction of the chromatin has been linked with an increased level of RAD51 at the locus of DSB and increased HR in constitutive heterochromatin (Chiolo et al Cell 2011). In contrast, depletion of dUTX results in less HR repair in facultative heterochromatin.

According to these observations, depletion of dUTX shouldn't promote the recruitment of RAD51 to polycomb bodies. Validating this hypothesis would further support their model. This can be achieved by investigating if dUTX impacts chromatin compaction and/or the recruitment of RAD51 in polycomb bodies.

We thank the reviewer for this insightful comment and have included several new experiments to address this point:

- 1) We indeed find that Rad51 does not engage with DSBs within polycomb bodies in control cells, as we only find Rad51 positive DSBs outside polycomb bodies (**Fig.5D**). Loss of dUtx indeed does not promote Rad51 recruitment inside polycomb bodies. In fact, we observe a reduction in Rad51 positive DSBs outside polycomb bodies following dUtx depletion (**Fig.5D**). This suggests that dUtx-mediated DSB movement is needed to load Rad51 outside polycomb bodies.
- 2) We find that the PRC1 complex member ph-p is reduced at DSBs inside polycomb bodies, already before these DSBs have moved (**Fig.S5C**). This reduction in ph-p levels at DSBs is rescued by the depletion of dUtx. Since the PRC1 complex (e.g. ph-p) is important for polycomb chromatin compaction, this suggests that DSBs inside facultative heterochromatin result in dUtx-mediated decompaction.

5. Further characterizing DNA repair events that occur inside and outside of the polycomb bodies (RPA, RAD51) is essential here given the different phenotypes associated with histone H3 demethylation in facultative and constitutive heterochromatin.

We agree and have included analyses of Rad51 localization inside and outside polycomb bodies (similar answer as to point 1): We find that in control cells, Rad51 solely localizes outside polycomb bodies and we almost never observe Rad51 at DSBs inside polycomb bodies (**Fig.5B-D**). When we deplete dUtx, we prevent movement of DSBs outside polycomb bodies and observe a significant reduction in Rad51 positive DSBs outside polycomb bodies (**Fig.5D**). This reinforces our conclusion that dUtx-mediated DSB movement is important to promote late HR steps (e.g. Rad51 loading).

In contrast to Rad51, ATRIP (binds to RPA-coated ssDNA) does bind to DSBs inside polycomb bodies and these ATRIP-covered DSBs move outside polycomb bodies (**Fig.4A-C**, figure similar to original submission). Loss of dUtx prevents movements of ATRIP-covered DSBs (**Fig.4C**).

Together, these two results indicate that early HR steps (end-resection, RPA binding) occur inside polycomb bodies, while late HR steps (Rad51 loading) only occur outside polycomb bodies, following dUtx-dependent DSB movement.

Reviewer 2.

Experiments/Results.

1. Shifting between the two I-SceI induction systems seems warranted and justified. However, the ecDHFR system is more or less constitutively active throughout all of development (at least during larval stages), whereas the heat shock system is induced at third instar larvae. Is there any data to suggest that DSB repair kinetics and factors may be different in 3rd instar larvae than breaks that are induced as early as 1st instar larvae? I do understand it may be experimentally impossible to induce the ecDHFR system in a more defined time point like the heat shock, but the authors may want to comment on this difference between the two systems.

We agree this is an important point. Recently, the lab of Jan LaRocque in fact performed experiments using the DR-*white* system (in euchromatin) in which they compared DSB repair pathway choice using heat-shock inducible I-SceI in several larval stages and found no differences in DSB repair in 1st, 2nd and 3rd instar larval stages (<https://pubmed.ncbi.nlm.nih.gov/38683763/>). We have included a reference to this work in our revised manuscript when introducing the two systems (**Fig.1D, E**).

2. It is interesting that a fair amount of DSBs are unresolved within the 60 minute window that it is observed (~16-50% depending on the induction method and what was being analyzed; almost 100% in IR-induced – S2C)). Do you know the kinetics to repair all events (>60 minutes)?

We have included two graphs showing the kinetics of Mu2 focus appearance to disappearance following I-SceI -DSB induction or IR-induced DSBs in facultative heterochromatin (**Fig.S4D, E**). It takes on average 30 minutes to resolve 50% of I-SceI induced Mu2 foci (**Fig.S4D**), and on average 80 minutes to resolve 50% of IR-induced Mu2 foci (**Fig.S4E**). These I-SceI timings are similar to what we have found previously using DSBs in eu- or constitutive heterochromatin (<https://pubmed.ncbi.nlm.nih.gov/27474442/>). IR induced DSBs show similar kinetics to what was previously found in mammalian cells, in which ~50% of IR-induced DSBs are repaired within 2hrs (see for example: <https://pubmed.ncbi.nlm.nih.gov/21317870/>).

We have included a brief discussion on the DSB kinetics in facultative heterochromatin, and the comparison with previous literature, when discussing our live imaging set-up (**Fig.2, Fig.S4D, E**).

3. Related, do you have a non-heat shock or non-TMP treated controls to determine whether some of the DSBs that are being analyzed are persisting from earlier points in development due to leaky expression of the I-SceI transgene?

We have now included these control experiments in the revised manuscript. For the non-heat shock control, we find $\pm 4\%$ of nuclei with 1 γ H2Av focus in the absence of heat-shock (**Fig.S1C**, with hsp.I-Sce, no heat-shock), which is comparable to γ H2Av levels without any hsp.I-SceI transgene present (**Fig.S1B**, no hsp.I-SceI samples).

For the non-TMP controls we performed DR-*white* repair product analyses in the absence of trimethoprim in the food, which yielded $<3\%$ of identified repair products (**Fig.S1D**).

Both controls suggest there is no, or little, leaky expression of the I-SceI transgene, since we find background γ H2Av/repair levels without I-SceI induction.

4. I also find it interesting that endogenous levels of Utx are important for survival in ATR-mutant background (S4H). Considering the deficiencies in repairing induced DSBs with siRNA KD of Utx, is there an increase in sensitivity to IR in the double mutants? dATR single mutants are hypersensitive to IR to begin with, so it may require lower doses to detect differences in the double mutants.

We thank the reviewer for this comment and have performed the experiment as suggested. Interestingly, we indeed find much stronger synthetic lethality when irradiating the double dUtx/ATR mutant with 10Gy γ -IR (0-4 days after egg laying). The irradiated double mutant has a 6% relative viability when compared to irradiated single dUtx mutants (**Fig.S8D**).

Methods/Statistics.

5. Throughout the description of the methods and results (i.e., Fig. 1A), the authors refer to NHEJ with insertions and deletions (indels) as "NHEJ". It is important to note that this system only detects NHEJ with indels (precise NHEJ is nicely noted in the figure). Related, "HR" should be defined some point as intrachromosomal HR (to contrast to inter-sister HR; line 574). This is helpful for those outside of the field that don't fully understand these nuances.

We apologize and agree with the reviewer that this wasn't clearly indicated in our original manuscript. We have included an improved description of the NHEJ and HR events that we can detect in both the main text as well as figure legend (**Fig.1A**).

For clarification; we can only detect an HR event using our DR-*white* system if an intact upstream *white* sequence is generated. This can in theory be generated following usage of the downstream *iwhite* sequence both intra-chromosomally as well as using the *iwhite* sequence on the sister chromatid. We apologize for not clearly stating this and have now

further clarified this in the text and the figure legend of **Fig.1A**.

6. Fig. S1D, the authors state that the relative proportion of HR vs. NHEJ is not different between lines. Where statistics completed on this?

We apologize for not stating this. We performed one-way ANOVA on the hsp-I-SceI samples (results now in **Fig. S1F**), which shows that DSB repair pathway choice between eu- and heterochromatic sites is not significantly different (similar to ecDHFR.I-SceI lines – **Fig.1I**). We added this information to the figure legends.

7. Fig. 2F, which fHet line was used?

This imaging analysis was performed following IR (5Gy γ -IR), therefore no fHet DR-*white* line was used here.

8. Fig. 3B, values are “ns”, but relative to what?

We apologize for not being clear. We have replaced **Fig.3B** with a figure showing the H3 levels before and after hsp.I-SceI induction (instead of the DSB/no DSB fold change) to make it clearer. ‘ns’ values now clearly indicate the statistical analyses performed on the ‘no hsp.I-SceI’ versus ‘hsp.I-SceI’ samples.

9. Fig. S4A, the “#” is defined as DSBs that were not resolved in the figure legend. However, the colors of the bars suggest some of them were (with white = resolved). For example, S4A, bottom two events suggest it was resolved based on white coloring at 20 minutes. But the “#” suggests they were not resolved. Is this because imaging was cut short for these events? If yes, maybe the white coloring that indicates “resolved” should be changed to “n.d.” (gray)? It’s also unclear why this figure, and none of the others, needed to highlight the unresolved DSBs with “#”.

We thank the reviewer for pointing this out. We have replaced the white coloring following ‘#’ with gray coloring to indeed indicate these DSBs were not resolved (now **Fig.S6A, B**). For these events, the imaging was indeed cut short. The reason we include ‘#’ in these imaging experiments, is because we only imaged for ~2 hours following IR, instead of ~16 hours when using the DR-*white*/ecDHFR-I-SceI system. We therefore have relatively many DSBs that we were not able to follow for a prolonged time period.

10. It wasn’t clear at first that each of the euchromatin sites and fHet sites were different lines. Adding this in the Fig. 1 legend or early into the manuscript (lines 87-90) would be

helpful.

We have now added a clearer description that these are different fly lines in the legend of Figure 1, as well as when describing the DR-*white* systems for the first time in the main text.

11. How did you irradiate the wing discs? X-ray, gamma? Please include in methods.

We apologize for not mentioning this. We irradiated using a Caesium-137 γ -ray source in an IBL 437C machine. We have now included this information in the methods.

Minor typos and citations.

12. Gene names should be italicized, although this may be journal-specific.

We changed this throughout the manuscript and in the legends.

13. Line 92, "NHEJ will generate" should be changed to "NHEJ may generate".

We changed this in the text.

14. Line 241, *Melanogaster* should be *melanogaster*

We changed this in the text.

15. High levels of H3K27me3 in Ubx is stated in several figure legends. This will be helpful to cite, unless it is common knowledge. Perhaps adding a citation in the methods on how the relative H2K27me3 levels were calculated?

We apologize and have added a reference to clarify our statement in the legend of **Fig.1C** (<https://pubmed.ncbi.nlm.nih.gov/27643538/>). We have also further clarified our ChIP normalization in the methods section.

Reviewer 3.

Major concerns.

1. The majority of repair in the DR-*white* reporter occurs via NHEJ (Figure 1I). How can this be explained since in this type of DR reporters (similar to that in the well-established DR-GFP reporter for homologous recombination) are mostly used to study homologous recombination?

The reviewer is right that these types of DR-reporters are often used to assess HR levels in specific contexts. Originally, this DR-*white* reporter was indeed designed by the lab of Jan LaRocque to be able to identify HR events *in vivo* by quantifying the number of flies with red eye color (intact upstream *white* gene), see <https://pubmed.ncbi.nlm.nih.gov/24368780/>.

However, since this development of the DR-*white* system, we have implemented PCR followed by Sanger DNA sequencing of the upstream *white* gene/I-SceI cut site in the DR-*white* reporter (see <https://pubmed.ncbi.nlm.nih.gov/27474442/>). This has allowed us to not only be able to quantify HR events, but also NHEJ events (small insertions and deletions). This was

achieved at the time by collaborating with the Bas van Steensel lab, who developed the TIDE algorithm (<https://pubmed.ncbi.nlm.nih.gov/25300484/>), which allows us to quantify specific HR/NHEJ events in Sanger sequences.

We have included more background on the development of the DR-*white* system(s), in the main text, the Figure 1 legend as well as methods to clarify these points.

2. Which NHEJ pathways act on DSBs in this reporter? This could be deduced from repair junction analysis and should be further validated by *Drosophila* Ku and PolQ knockdown for cNHEJ and MMEJ.

We agree with the reviewer that this is an important point. To address this question, we have now performed illumina sequencing of DR-*white* repair products and have collaborated with the lab of Marcel Tijsterman (Leiden UMC, the Netherlands) to analyze the different repair outcomes (**Fig.S3**). These analyses reveal that the majority of NHEJ events in facultative heterochromatin are due to canonical NHEJ repair with a small percentage ($\pm 3\%$ of all repair products, **Fig.S3B**) showing the presence of micro homologies in deletion products. We assigned these deletion products with >2 bp microhomologies as MMEJ. Because of this small proportion of identified MMEJ products, we decided to not cross our systems into Pol-theta (*mus308*) mutants.

However, to strengthen the conclusion that the identified NHEJ events in our PCR-TIDE analyses are due to canonical NHEJ, we depleted DmKu70 using RNAi and indeed find an increase in HR events (**Fig.S2C-D**), suggesting these are canonical NHEJ events. These results are in line with our previous characterizations of this DR-*white* reporter in euchromatin and constitutive heterochromatin (<https://pubmed.ncbi.nlm.nih.gov/27474442/>).

3. DSBs move away from heterochromatin for repair, which is dependent on dUtx and may be required for repair by homologous recombination. However, the majority of repair in the DR-*white* reporter occurs via NHEJ (see also previous point). Does NHEJ repair in heterochromatin also require DSB movement? This is particularly important given that the Van Steensel lab reported that MMEJ predominantly occurs in this chromatin context (Schep et al., Mol Cell, 2021). A more extensive analysis of DSB movement and repair pathway choice would increase the novelty of the work.

We agree with the reviewer that the question whether DSBs undergoing NHEJ also move is a very interesting point and we apologize for not discussing this in our initial submission.

Interestingly, we find the majority of DSBs in facultative heterochromatin to be dependent on NHEJ (**Fig.1I**), while we also observe movement of most, if not all, DSBs (**Fig.2C**). This could suggest that DSBs undergoing NHEJ also move outside polycomb bodies.

However, loss of dUtx results in defects in DSB movement (**Fig.4C, Fig.S6B**) as well as increased levels of NHEJ (**Fig.S7A**), suggesting that NHEJ does not *require* movement.

Finally, the low percentage of MMEJ repair events, as detected by illumina sequencing of repair products, was not affected by the absence of dUtx (**Fig.S7E**), indicating that MMEJ repair also occurs independently of DSB movement.

To strengthen the point that late HR steps in facultative heterochromatin do depend on dUtx activity and DSB movement, we have now included additional experiments where we analyze the Rad51 localization pattern inside and outside polycomb bodies (**Fig.5B-D**, see also our response to reviewer 1 - point 1). We find that in control cells, Rad51 solely localizes outside polycomb bodies and we almost never observe Rad51 at DSBs inside polycomb bodies. When we deplete dUtx, we prevent movement of DSBs outside polycomb bodies and observe a significant reduction in Rad51 positive DSBs outside polycomb bodies (**Fig.5D**). This reinforces our conclusion that dUtx-mediated DSB movement is important to promote later HR steps (e.g. Rad51 loading).

Together, our results suggest that NHEJ and MMEJ are insensitive to the specific DSB movements and can occur irrespective of DSB movement, while HR specifically depends on this dUtx-dependent DSB movement. We have now included these points into the discussion of our revised manuscript.

Finally, the van Steensel lab indeed finds relatively high levels of MMEJ in facultative heterochromatin when compared to NHEJ (<https://pubmed.ncbi.nlm.nih.gov/33848455/>). The assays developed in the van Steensel lab use Cas9-dependent DSB induction in human tumor cells, which differ from our analyses of I-SceI induced DSBs *in vivo* in flies. Moreover, their systems cannot assess HR levels (only MMEJ and NHEJ), while our DR-*white* reporter allows us to directly quantify HR. We therefore think there are multiple reasons for differences between their and our findings, which we have discussed in the discussion section.

4) The authors have studied only one heterochromatin mark on one side of the induced DSBs, and base all their conclusions with regard to DSB movement and repair on this analysis. Additional heterochromatin marks should be studied and the analysis of these marks should be extended to a larger region flanking either side of the induced DSBs.

We thank the reviewer for this comment and have included additional work to address these points:

- H3K27me3 and H2AK118 ubiquitination (H2AK119ub in mammals) are the only two canonical facultative heterochromatin marks described in literature thus far. Therefore, we have now also included ChIP data with an antibody that binds to H2AK118ub and also find loss of this histone modification upon single DSB induction at 3fH_2 (fHet3 in original submission) (**Fig.S5B**).
- We have also assessed the levels of the PRC1 complex member ph-p using an imaging approach and also find loss of this heterochromatic protein at DSBs inside polycomb bodies (before they have moved) (**Fig.S5C**).
- We have now included H3K27me3 ChIP-qPCR analyses using a primer set further away from the DSB site (3.1kb) and also find loss of H3K27me3 at this distance (**Fig.S5A**).
- In addition, as also mentioned in our response to reviewer 1 (point 1), we have now also included an orthogonal approach to monitor H3K27me3 levels at DSBs in polycomb bodies by employing imaging of a fluorescently tagged H3K27me3 mint-body. This mintbody also reveals a ~10% reduction in H3K27me3 signal at DSBs in facultative heterochromatin, which is prevented by dUtx depletion (**Fig.3E-G**).
- Finally, we now also include ChIP analysis of the constitutive heterochromatin mark H3K9me3 and, as expected, find low levels of this mark both before and after DSB induction in facultative heterochromatin (**Fig.S5E**).

5) Two out of three DR-white reporters in heterochromatin show an H3K27me3 reduction, one of which was studied and showed movement of DSBs for repair. This is too limited to draw firm conclusions about the necessity of movement for DSB repair in heterochromatin. Does the second heterochromatin DSB showing no reduction in H3K27me3 also move? If so, then movement is not dependent on loss of H3K27me3.

We thank the reviewer for this comment. We have been able to follow single DSBs in two of the three DR-*white* systems (2Fh_1 (fHet1) and 3Fh_2 (fHet3), see **Fig.S4A** and **Fig.2C** respectively). Unfortunately, we have been unable to recombine (cross) all live imaging components (ph-p-mCherry and eYFP-Mu2) into the DR-*white*/I-SceI flies on site 3Fh_1 (originally fHet2) to address the point whether these single DSBs also move.

However, to address this point in an alternative way (see also our response to reviewer 1 – point 1), we have now included an orthogonal assay to determine H3K27me3 levels at DSBs in polycomb bodies. In short, we employed an H3K27me3 mint-body, which is a fluorescently tagged H3K27me3 antibody developed by the lab of Hiroshi Kimura

(<https://pubmed.ncbi.nlm.nih.gov/21576221/>) (**Fig.3E-G**). This mint-body allows for the live imaging analysis of changes in H3K27me3 levels. Using this approach, we find a ~10% reduction in H3K27me3 levels at DSBs in polycomb bodies (following 5Gy IR), already before these DSBs move outside the heterochromatin domain (**Fig.3G**). This reduction in H3K27me3 levels at DSBs is completely prevented upon dUtx depletion, reinforcing our conclusion that dUtx is responsible for the observed loss of H3K27me3 at heterochromatic DSBs.

Moreover, our live imaging analyses of DSB movement upon 5Gy IR of cells shows the majority of DSBs move outside polycomb bodies and that this is defective upon dUtx depletion (**Fig.2F, 4C**).

Together, these results strongly suggest the dUtx-dependent H3K27me3 loss and DSB movement are not specific to two single DSB sites, but in fact reveal a more general mechanism at facultative heterochromatic DSB sites at both I-SceI as well as IR-induced DSBs.

6) RNAi against dUtx reduced homologous recombination in the *DR-white* at all three heterochromatic loci, while loss of H3K27me3 was only affected at two of these loci. This suggests that repair at these loci is not fully dependent on H3K27me3 loss, which would not be in line with the main conclusion of the work. Moreover, it also suggests that dUtx has roles in repair beyond its function in removing H3K27me3, yet how remains unclear. Does Utx modify other targets involved in repair?

We thank the reviewer for this comment. Indeed, we find that all *DR-white* integrations in facultative heterochromatin clearly depend on dUtx for HR repair (**Fig.5A**), while we did not observe evident H3K27me3 loss by ChIP at one locus (3fH_1 (fHet2)) (**Fig.3A**). The fact that we did not identify loss of H3K27me3 at this site could suggest that H3K27me3 demethylation has a very transient nature, potentially counteracted by histone methylation activities. Depending on the location within heterochromatin, these histone methylation activities might differ in activity.

However, we agree that the absence of H3K27me3 loss at this locus could in theory also suggest that dUtx has targets other than H3K27me3 that promote HR repair at this locus. Currently, dUtx has been described to be the only known demethylase for H3K27me3 in *Drosophila* (see <https://pubmed.ncbi.nlm.nih.gov/20212086/> and <https://pubmed.ncbi.nlm.nih.gov/18039863/>). There is indeed evidence that certain histone demethylases can target non-histone proteins, however, for dUtx this has not been described. However, we cannot formally exclude dUtx plays additional roles, besides H3K27me3

demethylation, at DSBs. We have included a brief discussion to accommodate this point raised by the reviewer in our revised discussion.

Finally, our new data provide additional evidence that loss of H3K27me3 signal at DSBs (**Fig.3E-G**), DSB movement (**Fig.4C**) and Rad51 binding outside polycomb bodies (**Fig.5D**) are all dependent on the presence of dUtx, strengthening our conclusion that dUtx-mediated H3K27me3 loss promotes DSB movement and HR.

7. The dominant type of repair in the DR-white reporter is NHEJ. Does RNAi against dUtx also impact this repair pathway and are there specific effects in euchromatin versus heterochromatin?

We apologize for not pointing this out clearly in our original submission. We find that dUtx RNAi leads to an increase in NHEJ repair products (**Fig.S7A**) specifically in facultative heterochromatin and has no effect on MMEJ levels (**Fig.S7E**). We hypothesize the increase in NHEJ to be due to a reduction in HR repair and may reflect changes in DSB repair pathway choice at the DSB site. We now comment on this in our revised discussion.

8) How specific is the single RNAi against dUtx? RNAi is known for its off-target effects, so a multiple RNAi approach or RNAi complementation approach should be considered to address this (particularly with regards to the two previous points).

We agree with the reviewer this is an important control. We find that a deletion mutant for dUtx (*dUtx[f01321]*, similar to the one used in Fig.S8D), also results in a decrease in HR levels (and a concomitant NHEJ increase) at facultative heterochromatin DSBs (**Fig.S7B**). Moreover, the dsRNA sequence used in all our cell culture experiments targets a different dUtx mRNA region than the dsRNA sequence used in our *in vivo* experiments, further strengthening the point that the effects we observe are not due to RNAi off-target effects.

9) H3K27me3 loss and DSB movement are required for homologous recombination, which depends on CtIP-dependent end-resection. But are H3K27me3 loss and/or DSB movement dependent on CtIP-dependent end-resection (or in other words are these processes affected by CtIP depletion)? If not, what does this mean to repair by HR versus NHEJ?

This is another interesting question raised by this reviewer. To address the reviewer's question, we have performed H3K27me3 ChIP analysis at single DSBs *in vivo* in the presence of DmCtIP knockdown. Our preliminary data suggest that loss of DmCtIP prevents the loss of H3K27me3 at these heterochromatic DSB sites (**Rebuttal Fig.1A**), while efficiently inducing

DSBs (γ H2Av ChIP **Rebuttal Fig.1B**). Moreover, DmCtIP depletion leads to defects in DSB movement outside polycomb bodies following IR of cell cultures (**Rebuttal Fig.1C, D**). This indeed suggests that end-resection could potentially initiate heterochromatin changes at DSBs and that there is an interplay between HR activities and chromatin changes at heterochromatic DSB sites, which we plan to further explore in our future projects.

REBUTTAL FIGURE 1

Rebuttal figure 1

A, B. H3K27me3 and γ H2Av ChIP analyses at indicated facultative heterochromatin DR-*white* sites +/- hsp.I-SceI in the presence of DmCtIP RNAi. **C, D** Analysis of DSB (eYFP-Mu2 foci) movement with respect to polycomb bodies (ph-p mCherry) following 5Gy IR in control **(C)** or DmCtIP depleted cells **(D)**. DSBs that appear in polycomb bodies were followed over time. Purple = inside polycomb body, green = outside polycomb bodies, grey = resolved.

Minor concerns.

10) The DR-white reporter resembles known reporters, most notably DR-GFP (initially developed by Maria Jasin's lab and used in the field for decades). Although extensive validation may not be needed, using CtIP knockdown for validation of homologous recombination of DSBs in the reporter is fairly limited. Also, CtIP is not uniquely involved in

homologous recombination and also plays a role in MMEJ. Knockdown of a core homologous recombination factors such as BRCA1, BRCA2, PALB2 or RAD51 would be required for further validation.

We agree that solely using DmCtIP depletion would not be sufficient to conclude HR repair can be quantified using this reporter (**Fig.S2B**). However, as also discussed in our response 1 and 2 to this reviewer, this DR-*white* reporter has been previously well characterized by others (<https://pubmed.ncbi.nlm.nih.gov/24368780/>) and us (<https://pubmed.ncbi.nlm.nih.gov/27474442/>). Moreover, our illumina sequencing analyses (**Fig.S3**) now reveal very low levels of MMEJ in repair products of this reporter, suggesting that the drop we see in HR repair products using DmCtIP knockdown indeed truly reflects HR loss.

As an additional control, we now also include DmRad51 knockdown in combination with the DR-*white* reporter in facultative heterochromatin and also find significantly reduced HR levels (**Fig.S2E**).

REVIEWERS' COMMENTS

Reviewer #1 (Remarks to the Author):

The authors are now providing additional proof to support their model. New orthogonal experiments show that dUtx directly impacts the levels of H3K37m3 within heterochromatin and the relocalization of the break outside the polycomb (as measured by the formation of RAD51 foci, which only occurs once the break has been moved out). These data strengthen the hypothesis that dUTX promotes DNA repair pathway choice in an H3K27me3-dependent manner.

Additional findings were provided in response to our comments, and our concerns have thus been addressed.

Reviewer #2 (Remarks to the Author):

The authors have sufficiently addressed my concerns. The manuscript has been significantly improved.

Reviewer #3 (Remarks to the Author):

The authors have thoroughly revised their manuscript by performing several additional experiments and by carefully clarifying several important points. I only have two minor points left (see below), which I think they can easily address. Other than that, I recommend publication of the manuscript.

- Why was the H2AK118ub ChIP-qPCR (Fig. S5B) not done at 3kb away from 3eu_1, to show that, similar to H3K27me (Fig. S5A), there is no signal and/or change in signal after DSB induction in a euchromatic region? This could be done easily.

- "We have also assessed the levels of the PRC1 complex member ph-p using an imaging approach and also find loss of this heterochromatic protein at DSBs inside polycomb bodies (before they have moved) (Fig.S5C)."

Images were not provided, making it impossible to grasp what was quantified in Fig. S5C. Please provide images.

There is loss of the PRC1 complex in heterochromatin at DSB, but what is the reference? Please explain.

We thank the reviewers for carefully assessing our manuscript after our rebuttal experiments. Below is our brief point-by-point response to the final comments of reviewer 3.

Reviewer 3.

1. Why was the H2AK118ub ChIP-qPCR (Fig. S5B) not done at 3kb away from 3eu_1, to show that, similar to H3K27me (Fig. S5A), there is no signal and/or change in signal after DSB induction in a euchromatic region?

We identified a decrease in the canonical facultative heterochromatic mark H3K27me3 specifically at DSBs induced in facultative heterochromatin (Fig. 3A). Euchromatic loci, as expected, are not enriched for H3K27me3 before DSB induction, and do not show any change in H3K27me3 upon DSB induction. This confirmed that the loss of heterochromatin marks is specific for DSBs in facultative heterochromatin.

The ChIPs for H2AK118ub (i.e. canonical facultative heterochromatin mark) were merely a control for the already observed specific loss of H3K27me3 levels at heterochromatic DSBs and we do not think assessing H2AK118ub at euchromatic DSBs is needed to draw our conclusion; i.e. that DSBs in facultative heterochromatin induce local loss of heterochromatin components.

Moreover, performing ChIPs using *Drosophila* larvae requires a substantial amount of time (starting crosses, collecting sufficient larvae) and would require ~1.5 months. We do not think the information we would acquire from this experiment warrants this time investment.

2. "We have also assessed the levels of the PRC1 complex member ph-p using an imaging approach and also find loss of this heterochromatic protein at DSBs inside polycomb bodies (before they have moved) (Fig.S5C)." Images were not provided, making it impossible to grasp what was quantified in Fig. S5C. Please provide images. We have now included representative images of an irradiation-induced γ H2Av focus in ph-p marked polycomb bodies (Fig.S5C). Additionally, we added a schematic of the quantification of ph-p levels at DSBs in polycomb bodies (Fig.S5D).

3. There is loss of the PRC1 complex in heterochromatin at DSB, but what is the reference? Please explain.

Our data, for the first time, shows loss of facultative heterochromatic marks at heterochromatic DSBs. This includes decreased levels of the facultative heterochromatic histone modifications H3K27me3 (Fig. 3A, E, F, G, S5A) and H2AK118Ub (Fig.S5B), as well as a loss of ph-p, a PRC1 complex member (Fig.S5C). As we are the first to report these findings, no existing references are available. We have slightly rephrased the text in the sentence referred to by the reviewer to clarify this point.